

# BioRT-Flux-PIHM v1.0: a watershed biogeochemical reactive transport model

Wei Zhi[1], Yuning Shi[2], Hang Wen[1], Leila Saberi[3], Gene-Hua Crystal Ng[3], Li Li[1,*]

[1] Department of Civil and Environmental Engineering, The Pennsylvania State University, State College, PA 16802, USA

[2] Department of Ecosystem Science and Management, The Pennsylvania State University, State College, PA 16802, USA

[3] Department of Earth Sciences, University of Minnesota, Twin Cities, MN 55455, USA

[*] Correspondence to lili@engr.psu.edu



## Abstract

Watersheds are the fundamental Earth surface functioning unit that connects the land to aquatic systems. Existing watershed-scale models typically have physics-based representation of hydrology process but often lack mechanism-based, multi-component representation of reaction thermodynamics and kinetics. This lack of watershed reactive transport models has limited our ability to understand and predict solute export and water quality, particularly under changing climate and anthropogenic conditions. Here we present a recently developed BioRT-Flux-PIHM (BFP) v1.0, a watershed-scale biogeochemical reactive transport model. Augmenting the previously developed RT-Flux-PIHM that integrates land-surface interactions, surface hydrology, and abiotic geochemical reactions (Bao et al., 2017, WRR), the new development enables the simulation of 1) biotic processes including plant uptake and microbe-mediated biogeochemical reactions that are relevant to the transformation of organic matter that involve carbon, nitrogen, and phosphorus; and 2) shallow and deep water partitioning to represent surface and groundwater interactions. The reactive transport part of the code has been verified against the widely used reactive transport code CrunchTope. BioRT-Flux-PIHM v1.0 has recently been applied to understand reactive transport processes in multiple watersheds across different climate, vegetation, and geology conditions. This paper introduces the governing equations and model structure of the code. It also demonstrates examples that simulate shallow and deep water interactions, and biogeochemical reactive transport relevant to nitrate and dissolved organic carbon (DOC). These examples were illustrated in two simulation modes of varying complexity. One is the spatially implicit mode that focuses on processes and average behavior of a watershed. Another is in a spatially explicit mode that includes details of topography, land cover, and soil property conditions. The spatially explicit mode can be used to understand the impacts of spatial structure and identify hot spots of biogeochemical reactions.

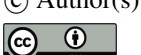



## 1. Introduction

Watersheds are the fundamental Earth surface units that receive and process water, mass, and energy (Li, 2019). Watershed processes include land-surface interactions that regulate evapotranspiration and discharge, and water partitioning between shallow soil lateral flow going into streams versus downward and recharge into the deeper subsurface (Figure 1). Complex biogeochemical interactions also occur between soil, water, roots, and microbe, dictating the $CO_2$ effluxes via soil respiration, export of soil weathering and biotransformation, and nutrient cycling (Fatichi et al., 2019;van der Velde et al., 2010).

These hydrological and biogeochemical processes determine how Earth surface responds to external forcings such as hydroclimatic drivers and human perturbations (van der Velde et al., 2014;Miller et al., 2020;Han et al., 2019;Steimke et al., 2018). Understanding these processes remain challenging due to the complex coupling of land surface, hydrology, and biogeochemical reactions (Kirchner, 2003). An example is the concentration-discharge (C-Q) relationships of solutes at stream and river outlets. These relationships encode integrated signature of Earth surface response to changes in hydrological conditions. Similar C-Q relationships have been observed for some solutes across watersheds under diverse geology and climate conditions (Godsey et al., 2009;Basu et al., 2010;Moatar et al., 2017;Zarnetske et al., 2018), whereas different solutes have shown contrasting patterns in the same watershed (Miller et al., 2017;Herndon et al., 2015;Zhi et al., 2019;Musolff et al., 2015). A general theory that can explain contrasting observations under diverse watershed characteristics and forcing conditions remains elusive. The lack of understanding of mechanisms that govern hydrological and biogeochemical interactions presents major roadblocks for forecasting water quality such that water issues such as eutrophication persist worldwide.

One of the challenges along these lines is the lack of modeling tools that mechanistically link hydrological and biogeochemical processes at the watershed scale. Model development has been advancing primarily within the disciplinary boundaries of hydrology and biogeochemistry (Li, 2019). Hydrology models that focus on solving for water storage and fluxes at the watershed scale and beyond (Fatichi et al., 2016), and



reactive transport models (RTMs) that center on aqueous and solid concentration
changes arising from transport and multi-component biogeochemical reactions typically
in "closed" groundwater systems without much interactions with "open" watersheds
directly receiving precipitation and sunlight (Steefel et al., 2015;Li et al., 2017b;Mayer et
al., 2002;MacQuarrie and Mayer, 2005). This comes along with a history of hydrologists
often trained as physicists studying fluid mechanics, and biogeochemists typically grow
up as geologists, chemists, or environmental engineers.

Recent works have shown some integration across these two lines. Examples

include HSPF (Hydrological Simulation Program – FORTRAN) (Filoso et al.,
2004;Laroche et al., 1996),  SWAT (Soil & Water Assessment Tool) (Gassman et al.,
2007;Lam et al., 2010;Moriasi et al., 2013;Neitsch et al., 2011), CATHY (Catchment
Hydrology) (Gatel et al., 2019;Scudeler et al., 2016), PAWS (Process-based Adaptive
Watershed Simulator) (Niu and Phanikumar, 2015;Qiu et al., 2019). These models have
relatively crude representations of solute leaching out of element bulk mass as part of the
solute export. These models do not represent kinetics and thermodynamics of multi-
component biogeochemical reactions typically done in reactive transport models (RTMs).
In filling in this model development need, recently we developed the watershed reactive
transport code RT-Flux-PIHM that integrates kinetics and thermodynamics of multi-
component geochemical reactions with the land-surface and hydrology model Flux-PIHM
(Bao et al., 2017). The geochemical reactions in RT-Flux-PIHM are abiotic, including
mineral dissolution and precipitation, aqueous and surface complexation, and ion
exchange reactions.

This manuscript introduces BioRT-Flux-PIHM (BFP) v1.0, augmented based on

RT-Flux-PIHM with two additions. One is the capability of simulating biotic processes
including plant uptake of nutrients, and microbe-mediated reactions in the soil. These soil
processes include the transformation of fresh and old organic matter, for example, soil
respiration that produces $CO_2$ and dissolved organic carbon (DOC), and nutrient cyclings
such as nitrification and denitrification. The other is the introduction of a deeper layer
below the shallow soil that enables the simulation of interactions of deep water and
shallow soil water flow (Figure 1). Here the deep water is loosely defined as the water
beyond the soil zone, typically in less weathered, fractured subsurface that harbors the


relatively old and slow-moving groundwater contributing to streams. This contrasts the
shallow water in highly permeable soils. Mounting evidence in recent years has shown
that the deeper water beyond the shallow soil interacts with streams, brings water with
distinct chemistry, sustains base flow in dry times, and buffers climate variability (Gurdak,
2017;Green, 2016;Taylor et al., 2013;Condon et al., 2013;Anyah et al., 2008;Maxwell et
al., 2011;Gleeson et al., 2015). They are therefore a fundamental component of the
hydrologic cycle and water budget. The groundwater-surface water interactions also
modulate land-atmospheric energy exchanges and soil moisture dynamics. Including the
deep water component therefore enables the simulation of such interactions and the
dynamics of water quality.

This paper introduces the governing equations, model structure and capabilities of
BFP. The biogeochemical reactive transport code has been verified against the widely
used reactive transport code CrunchTope (Supporting Information). We showcase the
model using two examples of varying complexity: one on nitrate processes run in a
spatially implicit mode; another on the production and transport of dissolved organic
carbon (DOC) in a spatially explicit mode with representation of spatial details. The source
code  and  the  examples  shown  here  are  hosted  in  the  Github  page
(https://github.com/PSUmodeling/BioRT-Flux-PIHM).

## 115 2. Model description

BioRT-Flux-PIHM integrates different processes in three modules (Figure 1). The
Flux module is for land-surface interaction processes including surface energy balance,
solar radiation, and evapotranspiration (ET) (Shi et al., 2013). The hydrology module
PIHM  simulates  water  processes  including  precipitation,  interception,  infiltration,
recharge, surface runoff, subsurface lateral flow, and deep water flow (Qu and Duffy,
2007). The BioRT module is for multi-component biogeochemical reactive transport
processes including microbe-mediated redox reactions (e.g., carbon decomposition and
nutrient transformation), ion exchange, aqueous and surface complexation, and mineral
dissolution and precipitation.

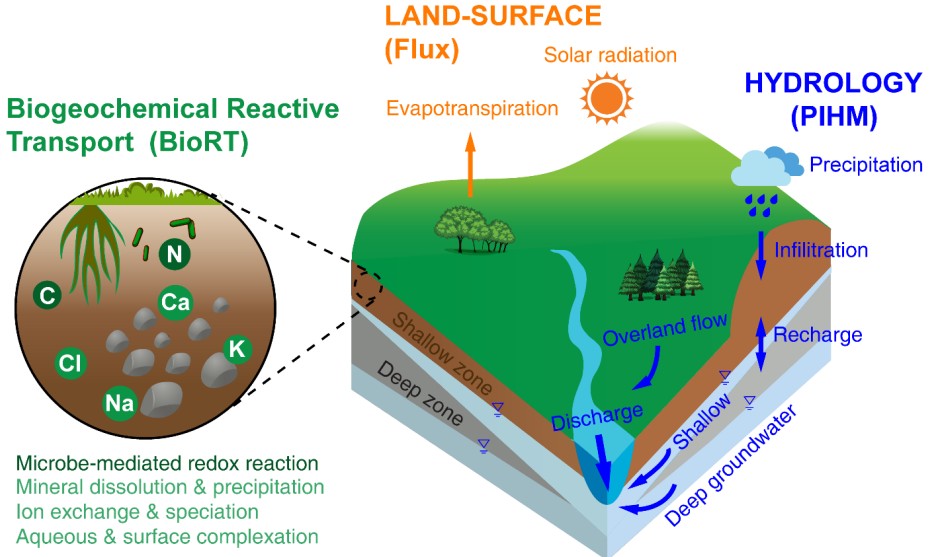


**Figure 1.** A conceptual diagram for processes at the watershed scale. This includes surface land interactions such as solar radiation, evapotranspiration); hydrological processes partitioning water between surface runoff, shallow soil water lateral flow, and deeper water entering the stream; and soil biogeochemical reactions including abiotic reactions (e.g., mineral dissolution and precipitation, ion exchange, surface complexations), and microbe-mediated reactions such as the transformation of carbon and nitrogen. These processes are represented in three modules: the Flux module for land-surface interaction processes, the PIHM module for hydrology processes, and the recently augmented BioRT module for soil biogeochemical reaction processes. Conceptually the shallow water zone includes the shallow subsurface such as soil and weathered zone that are more conductive to water flow (e.g., lateral flow or interflow). The deep zone refers to the less weathered, fractured zone that harbors the relatively old and slow flowing water that contributes to stream flow. Reactions can occur in both shallow and deep zones.

The land surface and hydrology modules are coupled to solve for temperature and water storage, from which water fluxes can be quantified for surface runoff, shallow and deep water fluxes. The BioRT module uses calculated temperature, water storage, and fluxes to simulate advection, diffusion / dispersion, and biogeochemical reactions in both shallow and deep zones. The reactions can be kinetics-controlled or thermodynamically controlled (e.g., ion exchange, surface complexation (sorption), and aqueous complexation). Users can define the type of reactions to be included and the form of

reaction kinetics in the input files. The output of BioRT includes the time series of aqueous
and solid concentrations in shallow and deep zones and in stream water.
The simulation domain can be discretized into prismatic grids based on
topography. Each grid is partitioned into surface and shallow and deep subsurface layers.
The shallow subsurface is loosely defined as the highly permeable subsurface that are
most conductive to water flow, contrasting the deep zone that is broadly defined as the
lower permeability zone beyond the shallow zone. In many places, this shallow zone is
the soil zone that is most conductive to water flow (e.g., lateral flow) and is very
responsive to hydroclimatic forcing. The deep subsurface zone is the less weathered,
fractured layer that harbors the relatively old and slow flowing water that contributes to
stream flow. Note that these definitions differ from those in the hydrology community,
which often refer to the shallow soil water flow as groundwater, in a way that distinguishes
from the surface runoff (Winter et al., 1998;Dingman, 2015;Todd and Mays, 2005). As
illustrated in Figure 1, both shallow and deep zone have unsaturated and saturated
layers, enabling the simulation of the "two water tables" (Brantley et al., 2017). These
transient water tables have been observed in catchments of Shale Hills, Garner Run, and
Cole Farm (Li et al., 2018;Brantley et al., 2018).

**3. Governing equations**

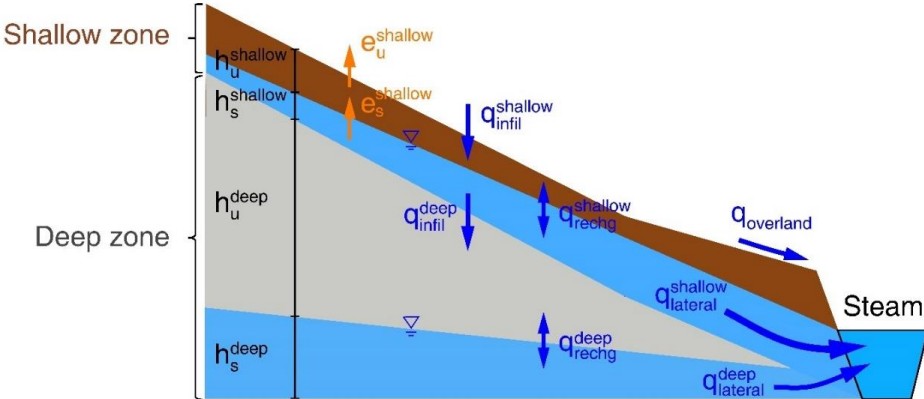


**Figure 2.** Hillslope view of the shallow and deep zones and relevant water flows. The symbol of
"h", "e", and "q" denotes water head, evapotranspiration, and water flow, respectively. The





subscript letter "u" and "s" refers to unsaturated and saturated layer, respectively. Detailed
equations are listed in the following sections.

## 3.1 The water equations in shallow and deep zones

As shown in Figure 2, in each prismatic element, the shallow zone has unsaturated
and saturated water storages and interactions between the two storages:
$$\theta^{shallow}\frac{dh_u^{shallow}}{dt} = q_{infil}^{shallow} - q_{rechg}^{shallow} - e_u^{shallow} \quad (1)$$

$$\theta^{shallow}\frac{dh_s^{shallow}}{dt} = q_{rechg}^{shallow} - q_{infil}^{deep} - e_s^{shallow} + \sum_{ij}^{1} q_{lateral\_ij}^{shallow} \quad (2)$$

Where $h_u^{shallow}$ and $h_s^{shallow}$ are the unsaturated and saturated water storage in the
shallow zone, respectively; $\theta^{shallow}$ is the shallow zone porosity; $q_{infil}^{shallow}$ and $q_{infil}^{deep}$ are
the shallow and deep infiltration from the surface to the shallow zone and from the shallow
to the deep zone, respectively; $q_{rechg}^{shallow}$ and $q_{rechg}^{deep}$ are the recharge from the unsaturated
layer to the saturated layer in the shallow and deep zones, respectively; $e_u^{shallow}$ and
$e_s^{shallow}$ are shallow evapotranspiration from the unsaturated and saturated layer (Shi,
2012), respectively; $q_{lateral\_ij}^{shallow}$ is the shallow normalized lateral flux in the saturated layer
from element $i$ to its neighbor $j$ ($\leq 3$).

Infiltration and recharge fluxes in the shallow zone for the elements $i$ are calculated
using the Richards equation, in which hydraulic water head $H$ (i.e., the summation of
water storage $h$ and elevation head $z$) and hydraulic conductivity $K$ determine the fluxes:
$$q_{infil}^{shallow} = AK_{infil}^{shallow}\frac{H_{sur} - H_u^{shallow}}{D_{inf}} \quad (3)$$

$$q_{rechg}^{shallow} = AK_{effV}^{shallow}\frac{H_u^{shallow} - H_s^{shallow}}{0.5D_{shallow}} \quad (4)$$

Where $A$ is the element area in the vertical direction; $D_{inf}$ and $D_{shallow}$ are the
thickness of infiltration (0.1 m) and shallow layer, respectively; $K_{infil}^{shallow}$ and $K_{effV}^{shallow}$ are
the infiltration and effective hydraulic conductivity in the vertical direction in the shallow





zone, respectively; $H_{sur}$ is the surface hydraulic water head ($= h_{sur} + z_{sur}$); $H_u^{shallow}$ and
$H_s^{shallow}$ are the shallow hydraulic water head in the unsaturated and saturated layer,
respectively. Shallow lateral flow in the saturated layer is calculated using Darcy's law:
$$q_{lateral\_ij}^{shallow} = A_{ij} K_{effH\_ij}^{shallow} \frac{(H_s^{shallow})_i - (H_s^{shallow})_j}{D_{ij}} \quad (5)$$

Where $A_{ij}$ is the projection area of the saturated layer between elements $i$ and $j$; $D_{ij}$ is
the distance between the centers of elements $i$ and $j$; $K_{effH\_ij}^{shallow}$ is the harmonic mean of
shallow effective hydraulic conductivity in the horizontal direction ($K_{effH}^{shallow}$) between
elements $i$ and $j$. The interaction between the shallow saturated zone and stream channel
also follows the Eq. (5), where the adjacent head is replaced by the level of the channel
water.

Similar to the shallow zone, the deep zone in each element $i$ can have unsaturated

and saturated storages, with unsaturated-saturated flow within $i$:
$$\theta^{deep} \frac{dh_u^{deep}}{dt} = q_{infil}^{deep} - q_{rechg}^{deep} \quad (6)$$

$$\theta^{deep} \frac{dh_s^{deep}}{dt} = q_{rechg}^{deep} + \sum_{j}^{1} q_{lateral\_ij}^{deep} \quad (7)$$

Where $h_u^{deep}$ and $h_s^{deep}$ are the unsaturated and saturated storages in the deep zone,
respectively; $\theta^{deep}$ is the deep zone porosity; $q_{rechg}^{deep}$ is the deep recharge flux from the
unsaturated layer to the saturated layer; $q_{lateral\_ij}^{deep}$ is the deep normalized lateral flux from
element $i$ to its neighbor $j$ ($\leq 3$).

Deep lateral flow is calculated using Darcy's law:

$$q_{lateral\_ij}^{deep} = A_{ij} K_{effH\_ij}^{deep} \frac{(H_s^{deep})_i - (H_s^{deep})_j}{D_{ij}} \quad (8)$$

Where $H_s^{deep}$ is the deep hydraulic water head; $K_{effH\_ij}^{deep}$ is the harmonic mean of the deep
effective hydraulic conductivity in the horizontal direction ($K_{effH}^{deep}$) between elements $i$ and
$j$.





Deep infiltration and recharge fluxes are similarly calculated using the Richards
equation as in the shallow zone:

$$q_{infil}^{deep} = AK_{infil}^{deep} \frac{H_s^{shallow} - H_u^{deep}}{0.5\left[H_s^{shallow} + \left(D^{deep} - H_s^{deep}\right)\right]} \quad (9)$$


$$q_{rechg}^{deep} = AK_{effV}^{deep} \frac{H_u^{deep} - H_s^{deep}}{0.5D^{deep}} \quad (10)$$

Where $K_{infil}^{deep}$ is the hydraulic conductivity of infiltration from the shallow zone to the deep
zone; $D^{deep}$ is the thickness of the deep zone; $K_{effV}^{deep}$ is the effective hydraulic conductivity
in the vertical direction of the deep zone.
The deep groundwater can also come from regional groundwater aquifers, which
can set up as an influx from the boundary of the domain. Deep groundwater interacts with
river channel via the shallow zone. When the level of deep groundwater is higher than the
depth to the deep zone, i.e., the shallow transient groundwater and the deep groundwater
are connected, the deep groundwater can flow into the transient saturated layer in the
shallow zone:

$$q_{infil}^{deep} = -AK_{satV}^{deep} \quad (11)$$

Where $K_{satV}^{deep}$ is the saturated hydraulic conductivity in the vertical direction of the deep
zone.
**Macropores.** Macropores, including roots and soil cracks are omnipresent in soils.
Macropore flows can be simulated in the model to account for rapid water flows in the
shallow zone (Shi et al., 2013). Macropore properties include depth ($D_{mac}$) and macropore
vertical and horizontal area fraction ($f_{macV}$ and $f_{macH}$), and vertical and horizontal
hydraulic conductivity ($K_{macV}^{shallow}$ and $K_{macH}^{shallow}$). The macropore depth differs from the
rooting depth, which specifies the maximum depth of transpiration. By default $K_{macV}^{shallow}$
and $K_{macH}^{shallow}$ are 100 and 1,000 times of the infiltration hydraulic conductivity ($K_{infil}^{shallow}$)
and shallow horizontal hydraulic conductivity ($K_{satH}^{shallow}$), respectively, and can be changed
during calibration. Taking both soil and macropore properties into account, the effective



hydraulic conductivity of the subsurface is calculated as the weighted average of the
macropore and the shallow soil matrix within the macropore depth (Eq. (12) and (13)).
$$K_{effV}^{shallow} = f_{macH}K_{macV}^{shallow} + (1 - f_{macH})K_{satV}^{shallow} \quad (12)$$

$$K_{effH}^{shallow} = f_{macV}K_{macH}^{shallow} + (1 - f_{macV})K_{satH}^{shallow} \quad (13)$$


## 3.2 Biogeochemical reactive transport equations

The governing equation for an arbitrary solute $m$ in grid $i$ is as follows (Bao et al.,
2017):

$$V_i\frac{d(S_{w,i}\theta_i C_{m,i})}{dt} = \sum_{j=N_{i,1}}^{N_{i,x}} \left( A_{ij}D_{ij}\frac{C_{m,j} - C_{m,i}}{I_{ij}} - q_{ij}C_{m,j} \right) + R_{m,i} \quad (14)$$

Where $V_i$ is the total volume of grid $i$ (solid + liquid volume), m$^3$; $S_{w,i}$ is soil water
saturation, m$^3$ water/m$^3$ pore space; $\theta_i$ is porosity, m$^3$ pore space/m$^3$ total volume; $C_{m,i}$ is
the aqueous concentration of species $m$, mol/m$^3$ water; $N_{i,x}$ is the index of elements
sharing surfaces; the value of $x$ is 2 for the unsaturated zone (infiltration, recharge) and
4 for the saturated zone (recharge plus three lateral flow directions), respectively; $A_{ij}$ is
the interface area (m$^2$) shared by $i$ and its neighbor grid $j$; $D_{ij}$ is the combined
dispersion/diffusion coefficient (m$^2$/s) normal to the shared surface $A_{ij}$; $I_{ij}$ is the distance
between the center of $i$ and its neighbor elements $j$; $q_{ij}$ is the flow rate across $A_{ij}$, m$^3$/s;
$R_m$ is the total rate of kinetically controlled reactions that involve species $m$, mol/s.
Various types of reaction occur in the subsurface (Fatichi et al., 2019). Generally
speaking, shallow soils contain more weathered materials and organic matters (OM)
including roots, leaves, and microbe. In contrast, deeper zones are less weathered and contain
much less organic matter. SOM can decompose partially into organic molecules that
dissolve in water (Wieder et al., 2015), i.e., DOC, or oxidize completely into $CO_2$ gas or
dissolved inorganic carbon (DIC). With coexisting divalent cations (e.g., Ca, Mg), DIC can
also precipitate and become carbonate minerals. Hence soil C decomposition can release
$CO_2$ back into the atmosphere and changes $CO_2$ level (Davidson, 2006), or releases DOC





and DOM to surface water. These processes occur in soils and also as dissolved carbon
transport laterally to streams.

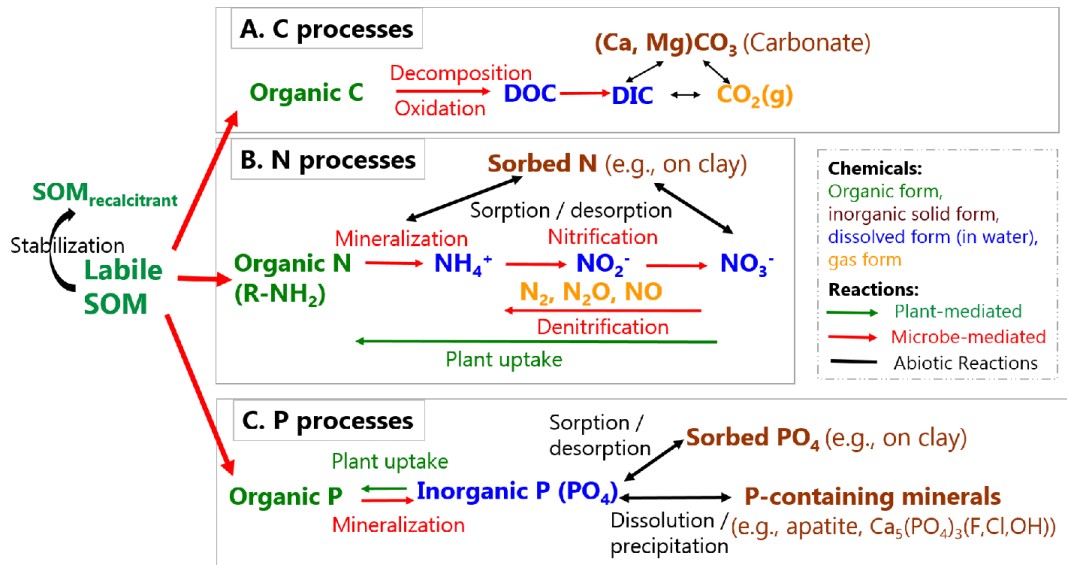


**Figure 3.** Various types of biotic and abiotic reactions relevant to the transformation of soil organic
matter (SOM). It can become stabilized through sorption on clay and separation from reactants.
It can also decompose into inorganic forms, transitioning between different phases (adopted from
Li (2019), permission with Mineralogical Society of America).

Shown in Figure 3, SOM decomposition releases organic nitrogen ($R-NH_2$), which

can further react to become ammonia, and other forms in between ($N_2$, $N_2O$, NO, $N_2O_3$
($NO_2^-$), $NO_2$). Some of the gaseous forms emit back to the atmosphere (Saha et al.,
2017;Maavara et al., 2018). Denitrification requires anoxic conditions and does not occur
as much in shallow soils owing to the pervasive presence of $O_2$ (Sebestyen et al., 2019);
it can become prevalent however under extremely wet conditions and in $O_2$-depleted
groundwater systems. In soils, P can be in organic form (e.g., leaves), sorbed (on fine
soil particles), dissolved in water, or in solid forms as P-containing minerals (Figure 3).
The transformation between different forms occurs through various bio-mediated or
abiotic reactions. The most abundant P-containing mineral is apatite $Ca_5(PO_4)_3$(F, Cl,
OH). Once liberated via rock dissolution, P is mostly locked in organisms. It is barely
soluble so it binds on and transports together with soil particles in the form of





orthophosphate or pyro-diphosphate. Overall, these reactions are a combination of biotic
and abiotic reactions.
BioRT can simulate biotic reactions including microbe-mediated reactions and plant
uptake, in addition to the abiotic reactions such as mineral dissolution and surface complexation
or ion exchange that have been introduced by Bao et al. (2017). Here we focus on the discussion
of a few representative microbe-mediated reactions.

**Microbe-mediated reaction kinetics.** SOM is often conceptualized and modeled as
pools with different decomposition rates and turnover times (Ostle et al., 2009;Thornton
et al., 2009). An extensively used three-pool model includes a readily degradable (labile)
pool with residence times less than five years; a slowly degrading pool with residence
times of decades; and a relatively stable pool, with residence times between $10^3 - 10^5$
years (Trumbore et al., 1995;Trumbore, 1993;Marin-Spiotta et al., 2009). The kinetics of
microbe-mediated reactions can be described by the general dual Monod rate law,
reflecting the need for both electron donor and acceptor in these reactions (Monod, 1949):
$$r = \mu_{max} C_{C_5H_7O_2N} \frac{C_D}{K_{m,D} + C_D} \frac{C_A}{K_{m,A} + C_A} \quad (15)$$

Here $\mu_{max}$ is the rate constant (mol/time/microbe cell), $C_{C_5H_7O_2N}$ is the concentration of
microorganisms (microbe cells/L$^3$), $C_D$ and $C_A$ are the concentrations of electron donor
and acceptor (mol/L$^3$), respectively. The $K_{m,D}$ and $K_{m,A}$ are the half-saturation coefficients
of the electron donor and acceptors (mol/m$^3$), respectively; they are the concentrations at
which half of the maximum rates are reached for the electron donor and acceptor,
respectively. If an electron donor or acceptor is not limiting, it means that $C_D \gg K_{m,D}$ or
$C_A \gg K_{m,A}$, so that the term $\frac{C_D}{K_{m,D}+C_D}$ or $\frac{C_A}{K_{m,A}+C_A}$ is essentially 1, lending to a rate that only
depends on the abundance of microorganisms or one of the chemicals.
In natural subsurface where multiple electron acceptors coexist, the
biogeochemical redox ladder dictates the sequence of redox reactions. That is, aerobic
oxidation occurs before denitrification, which in turn occurs before iron reduction.
Inhibition terms are used to account for the sequence of redox reactions as follows:



$$r = \mu_{max} C_{C_5H_7O_2N} \frac{C_D}{K_{m,D} + C_D} \frac{C_A}{K_{m,A} + C_A} \frac{K_{I,H}}{K_{I,H} + C_H} \quad (16)$$

Here $K_{I,H}$ is the inhibition coefficient for the inhibiting chemical $H$. The inhibition term is 1
(not inhibiting) only when $C_H \ll K_{I,H}$. In a system where oxygen and nitrate coexist, which
is common in agriculture lands, aerobic oxidation occurs first before denitrification. The
denitrification rates can be represented by:
$$r_{NO_3^-} = \mu_{max} C_{C_5H_7O_2N} \frac{C_D}{K_{m,D} + C_D} \frac{C_{NO_3^-}}{K_{m,A} + C_{NO_3^-}} \frac{K_{I,O_2}}{K_{I,O_2} + C_{O_2}} \quad (17)$$

Here $C_{NO_3^-}$ is the concentration of nitrate, $K_{I,O_2}$ is the inhibition coefficient of $O_2$, or the $O_2$
concentration at which it inhibits the reduction of nitrate. This rate law ensures that
denitrification kicks in substantially only when $O_2$ is depleted to $C_{O_2} \ll K_{I,O_2}$, such that the
term $\frac{K_{I,O_2}}{K_{I,O_2} + C_{O_2}}$ approaches 1.0. If there exists an electron acceptor that is lower in the
redox ladder than nitrate, multiple inhibition terms are needed. For example, for iron
oxide, we write the following:
$$r_{Fe(OH)_3} = \mu_{max} C_{C_5H_7O_2N} \frac{C_D}{K_{m,D} + C_D} \frac{C_{Fe(OH)_3}}{K_{m,Fe(OH)_3} + C_{Fe(OH)_3}} \frac{K_{I,O_2}}{K_{I,O_2} + C_{O_2}} \frac{K_{I,NO_3^-}}{K_{I,NO_3^-} + C_{NO_3^-}} \quad (18)$$

Here $K_{I,NO_3^-}$ is the NO3- concentration above which it inhibits iron reduction. The additional
nitrate inhibition term means that iron reduction occurs at significant rates only when both
oxygen and nitrate are low compared to their corresponding inhibition coefficients.

**Rates in natural soils.** The dual-Monod and inhibition terms are important under
conditions where electron donors and acceptors are limited. In shallow soil, $O_2$ is
prevalent except under wet conditions with little pore space for air. Anoxic conditions can
also develop in local environments such as dead-end pores where water is saturated for
a long time and not easily flows out. Under conditions organic carbon and $O_2$ are
abundant, the SOM rate law is simplified to the following form assuming microorganism
concentrations are relatively constant:
$$r_{SOM} = \mu_{max} A f(T) f(S_w) \quad (19)$$



Where the reaction rate $r_{SOM}$ (mol /t) now depends on $\mu_{max}$ (mol/m²/t), the lumped
surface area $A$ (m²) as an approximation of SOM content and biomass abundance, and
$f(T)$ and $f(S_w)$ describe its temperature and soil moisture dependence, respectively. For
temperature dependence, a $Q_{10}$-based form (Friedlingstein et al., 2006;Hararuk et al.,
2015) is commonly used:  $f(T) = Q_{10}^{|T-20|/10}$, where $Q_{10}$ is the relative increase in
reaction rates when temperature increases by 10 °C (Davidson and Janssens, 2006). The
$f(S_w)$ accounts for the nonlinear dependence of rates on soil moisture. A simple form
of $f(S_w) = (S_w)^\varepsilon$ where $\varepsilon$ is the saturation exponent (a typical $\varepsilon$ value is 2, with a range
between 1.5 and 2.5) is often used. More complex forms of $f(S_w)$ considering both water
limitation under dry conditions and $O_2$ limitation under wet conditions have been proposed
(Yan et al., 2018). It has also been suggested that the decomposition depends strongly
on the depth distribution of SOM (Seibert et al., 2009), which is sometime accounted with
an additional depth function:
$$r_{SOM} = \mu_{max} A f(T) f(S_w) f(Z_w) \quad (20)$$
where $Z_w$ is the water table depth (m). An example is $f(Z_w) = \exp\left(-\dfrac{Z_w}{b_m}\right)$ (Weiler and
McDonnell, 2006;Ottoy et al., 2016;Bai et al., 2016). Here $b_m$ is the declining coefficient
describing the gradient of SOM content over depth.

**4. Numerical scheme and model verification**
**Numerical scheme.** The local system of differential equations for water storages [e.g.,
Eq. (1), (2), (6), and (7)] on each control volume are combined into a global system of
ordinary differential equations (ODEs) and solved in CVODE, a numerical ODE solver in
the SUite of Nonlinear and Differential / ALgebraic equation Solvers (SUNDIALS)
(Hindmarsh et al., 2005). CVODE is a numerically efficient solver for ODE systems. It
uses the backward difference formula (BDF) with adaptive time steps and method order
varying between 1 and 5. At each iteration step, the solver evaluates the local error, which
is required to satisfy convergence tolerance conditions set by the users. The internal time
step is reduced and the method order is adjusted in response to the stiffness of ODEs if
the non-convergence occurred. For example, the solver time steps become smaller after





heavy precipitation events to address the rapid change of surface and subsurface water storages. The adaptive time stepping and order adjustment scheme make CVODE an accurate and stable solver.

**Model verification.** The BioRT module had been verified against CrunchTope under a variety of transport and reaction conditions at a range of reaction complexity levels (Supporting Information, Figure S1 – S7). CrunchTope is a widely used subsurface reactive transport model (Steefel and Lasaga, 1994;Steefel et al., 2015), and is often used as a benchmark to verify other reactive transport models. Verification is performed under simplified hydrological conditions with 1-D column and constant flow rates such that it focuses on biogeochemical reactive transport processes such as advection, diffusion, dispersion, and biogeochemical reactions. Specifically, three cases of soil phosphorus, carbon, and nitrogen were verified for temporal evolution and spatial pattern of relevant solute concentrations (Figure S1 – S7). The soil phosphorus case, which involves geochemically kinetic and thermodynamic processes (i.e., apatite dissolution and phosphorous speciation), was first tested for solution accuracy of the bulk code that was inherited from the original RT-Flux-PIHM. Soil carbon and nitrogen processes that involve microbially driven processes, such as soil carbon decomposition and mineralization, nitrification and denitrification, were further verified for solution accuracy of the augmented BioRT module.

**4 Model setup and data needs**

**4.1 Model structure and input/output**

Flux-PIHM sets up the domain based on watershed characteristics including topography, hydrography, land cover, and shallow and deep zone properties (Figure 4). It takes in meteorological forcing and solves for water storages and soil temperature. BioRT takes in the model output of water and temperature, and drives the simulation for biogeochemical reactive transport. At the time scale of months to years that are typical for BioRT-Flux-PIHM simulations, the alteration in solid phase properties due to reactions is considered negligible and does not change hydrological parameters.



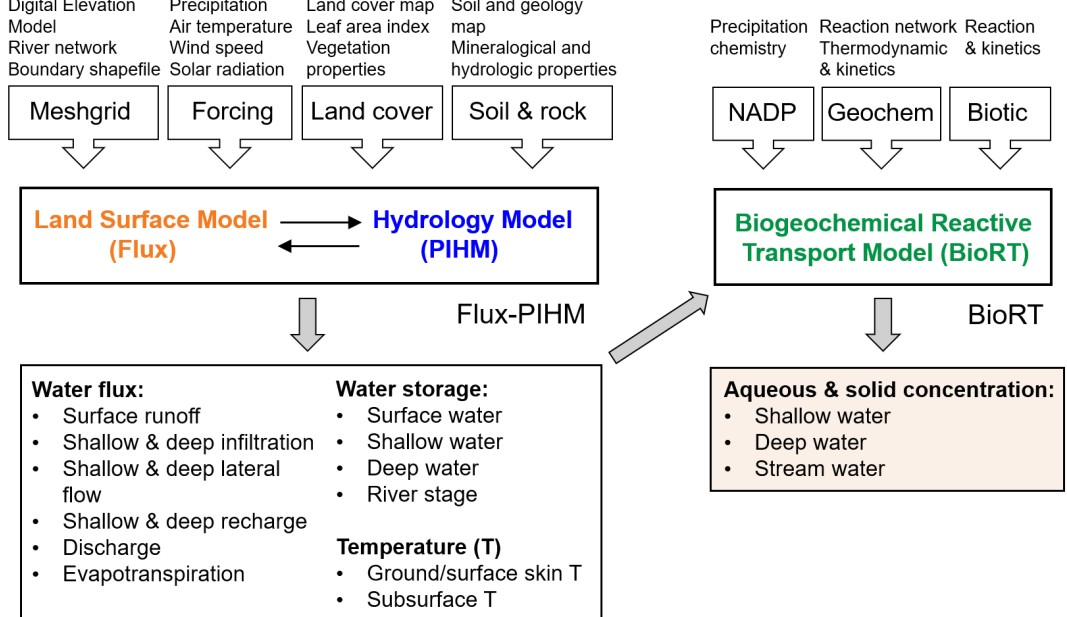

**Figure 4.** Model structure, input, and output of BioRT-Flux-PIHM. The Flux-PIHM takes in watershed characteristics including topography (digital elevation model, DEM), land cover, shallow and deep zone properties, and meteorological forcing and then solves for water storage and fluxes, and ground and soil temperature. Water- and temperature-related information from Flux-PIHM with additional inputs such as precipitation chemistry and shallow and deep water chemistry and biogeochemical kinetics parameters are then provided for the BioRT module, which eventually outputs aqueous and solid concentration for the shallow and deep zone, and stream water. NADP stands for the National Atmospheric Deposition Program.

Most model inputs such as elevation, land cover, soil and geology map can be obtained from the data portal of Geospatial Data Gateway (https://datagateway.nrcs.usda.gov). The meteorological forcing data can be downloaded from the North American Land Data Assimilation Systems Phase 2 (NLDAS-2, https://ldas.gsfc.nasa.gov/nldas/v2/forcing). The vegetation forcing, i.e., Leaf Area Index (LAI), is from the Moderate Resolution Imaging Spectroradiometer (https://modis.gsfc.nasa.gov/data). Other vegetation properties associated with land cover (e.g., shading fraction, rooting depth) are adopted from the Noah vegetation parameter table embedded in the Weather Research and Forecasting model (WRF;



Skamarock and Klemp (2019)). Local measurements from meteorological stations and
field campaigns (e.g., land cover, soil, geology) can also be used in the model. Another
data source for the model input is the HydroTerre (http://www.hydroterre.psu.edu/), where
users can obtain geospatial data (Leonard and Duffy, 2013). The form of microbial
reaction rate laws, when it includes full Monod form, or only with temperature and soil
moisture dependence, can be defined in the input files. Additional inputs include initial
water and solid phase chemistry, description of solutes and biogeochemical reactions,
and kinetics and thermodynamics of reactions from a geochemical database. The model
outputs include aqueous and solid concentrations of shallow and deep zone and stream
water.

**430   4.2 Model setup: from simple, spatially implicit to complex, spatially explicit**

**431   domains**

The model domain can be set up using PIHM-GIS
(http://www.pihm.psu.edu/pihmgis_home.html), a standalone GIS interface for watershed
delineation, domain decomposition, and parameter assignment (Bhatt et al., 2014). The
domain can be set up at different spatial resolutions with a different number of grids. A
simple domain can be set up with only two land grids representing two sides of a
watershed connected by one river cell (Figure 5). This setup uses averaged properties
without considering spatial details. This type of model setup requires less spatial data, is
computationally inexpensive, and is relatively easy to set up. It can be used to assess the
average dynamics of the water and solute dynamics and focus on the interactions among
processes without concerning spatial details. It can also be used as a relatively easy
starter for educational purposes before students jump into complex domains.
Alternatively, a complex domain can be set up using many grids with explicit
representation of spatial details. It requires much more data and is computationally
expensive but can be used to identify "hot spots" of biogeochemical reactions.





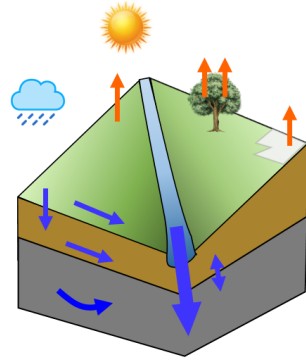

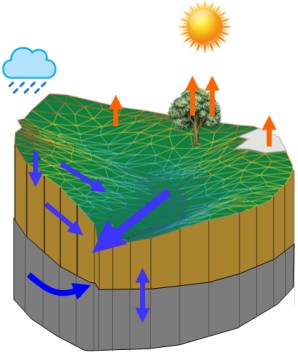

**Figure 5.** Two extreme model domain setups: a simple domain with two land cells representing two hillslopes connected by one river cell, versus a complex domain with hundreds of land cells. An intermediate number of grids can also set up the domain, depending on watershed heterogeneity, data availability, and desired spatial resolution.

## 5. Model applications

The original RT-Flux-PIHM has been applied to understand the processes related to geogenic solutes of Cl and Mg at the Shale Hills watershed and for geogenic Na at the Volcán Chimborazo watershed (Table 1). The new BioRT-Flux-PIHM has been demonstrated for understanding the dynamics of DOC, nitrate, and Na at Shale Hills and Coal Creek. This section presents some new model features using two examples: one with a simple, spatially implicit domain, and another with a complex, spatially explicit domain.

**Table 1.** Model applications with different biogeochemical reactions

| Watershed (location) | Size (km$^2$) | Model domain | Modeled solutes | Reaction network (Kinetic rate law: 1, TST; 2, Monod based; 3, plant uptake rate) | Reference |
| --- | --- | --- | --- | --- | --- |





| | | Complex (535 grids) | Cl, Mg | • Chlorite dissolution[1] <br>• Illite dissolution[1] <br>• Carbonate dissolution & precipitation[1] <br>• Cation exchange | Bao et al., 2017; Li et al., 2017 |
| Shale Hills (PA, USA) | 0.08 | Complex (535 grids) | DOC | • SOC decomposition[2] <br>• DOC sorption | Wen et al., 2020 |
| | | Simple (2 grids) | NO$_3^-$ | • Soil N leaching[2] <br>• Denitrification[2] <br>• Plant nitrate uptake[3] | This work |
| Coal Creek (CO, USA) | 53 | Simple (2 grids) | DOC, Na | • SOC decomposition[2] <br>• DOC sorption <br>• Albite dissolution[1] | Zhi et al., 2019 |
| Volcán Chimborazo (Ecuador) | | Complex (160 grids) | Cl, Na, Ca, Mg, SiO$_2$ | • Albite dissolution[1] <br>• Diopside dissolution[1] | Leila et al., 2020 (under review) |

Note: Transition State Theory (TST) is a classic kinetic rate law for mineral dissolution and
precipitation (Brantley et al., 2008); Monod based rate law with environmental dependency (i.e.,
soil temperature and soil moisture) is widely used for microbial driven reactions; plant nitrate
uptake depends on nitrate availability, environmental dependency, and rooting depth. Monod
based and plant nitrate uptake rate law are detailed in the following section of 5.1.1.

Here we present two examples of different processes in the Susquehanna Shale
Hills Critical Zone Observatory (SSHCZO), a small headwater watershed (0.08 km$^2$) in
central Pennsylvania, USA. The mean annual precipitation is approximately 1,070 mm
and the mean annual temperature is 10 °C. Extensive field measurements have been
conducted to characterize the topography, vegetation, and bedrock and soil properties
(Brantley et al., 2018). Soil carbon storage and respiration and nitrogen budget and fluxes
have been detailed studied (Andrews et al., 2011;Hasenmueller et al., 2015;Shi et al.,
2018;Hodges et al., 2019;Weitzman and Kaye, 2018). Modeling work has also been
conducted to understand hydrological dynamics (Shi et al., 2013), transport of a non-
reactive tracer Cl and the soil and rock weathering Cl and Mg (Bao et al., 2017;Li et al.,
2017a).






### 5.1 Hydrology Example: Shallow and deep water interactions

The model was set up using two land grids and one river grid, represented by the
averaged land cover, soil and rock properties based on previous work (Shi et al.,
2013;Kuntz et al., 2011). Specifically, the model assumed a Weikert soil, the dominant
soil type at Shale Hills (Shi et al., 2013). The porosity of the deep zone was set to 1/10 of
the shallow soil porosity based on measurements of the deep subsurface (Brantley et al.,
2018;Kuntz et al., 2011). Stream discharge and ET observations were used to calibrate
hydrological parameters (Figure S9). Groundwater ($Q_G$) from the deep layer was
constrained by previous work (Li et al., 2017a) and the nitrate concentration-discharge
(C-Q) observations. Important land surface and hydrological parameters are summarized
in Table S7.
**Water budget.** The model reproduced the seasonal dynamics of discharge and ET
(Figure S9), with daily Nash-Sutcliffe efficiency (NSE) of 0.56 and 0.66, respectively.
Precipitation occurs throughout the year while the discharge was responsive to a few big
storm events in spring and fall. The ET peaked during the summer due to higher solar
radiation and higher temperature while declined in the fall and winter. The runoff ratio was
0.46, suggesting 46% of precipitation was discharged through the stream while the
remaining 54% contributed to ET. A breakdown analysis suggests at the annual scale,
the shallow lateral flow ($Q_L$, 87% of Q) dominated discharge, followed by the deeper
groundwater flow ($Q_G$, 9.3%), and the surface runoff ($Q_S$, 4.2%). The $Q_G$ was essential in
maintaining discharge during dry time, especially in the summer.
**Controls of deep water.** In a headwater watershed like Shale Hills where the deep
groundwater is most likely sourced from recharge, the deep groundwater contribution to
the stream was primarily controlled by hydraulic conductivity ($K_{satH}$) contrast between the
deep and shallow zone (i.e., $K_{satH}^{deep} / K_{satH}^{shallow}$). Because the $K_{satH}$ contrast determined the
partitioning of infiltrating water between the shallow lateral flow and the downward
recharge to the deep zone and then deep groundwater flow. Two cases of high (red) and
low (blue) $K_{satH}^{deep}$ were set up to showcase the $K_{satH}$ contrast control on deep groundwater



(Figure 6a). By changing the deep zone $K_{satH}^{deep}$ from 2.6 to 0.22 (m/d), about 38% and 3.1%
of the shallow zone $K_{satH}^{shallow}$, annual deep groundwater ($Q_G$) contribution to discharge (Q)
decreased from 26% to 5.2%, respectively. It is also noticeable that there was minimal
change between discharge (i.e., solid lines in Figure 6a) as the deep zone $K_{satH}^{deep}$ does not
affect shallow water partitioning for infiltrating water and discharge. This new hydrology
feature enables the exploration of the interaction between deep groundwater and surface
water. These features can be used to understand watersheds of different subsurface
structures and with deep water mostly from recharge. In addition, they can be used to
explore large watersheds of higher stream order with a large proportion of deep water
coming from nearby regional aquifers.

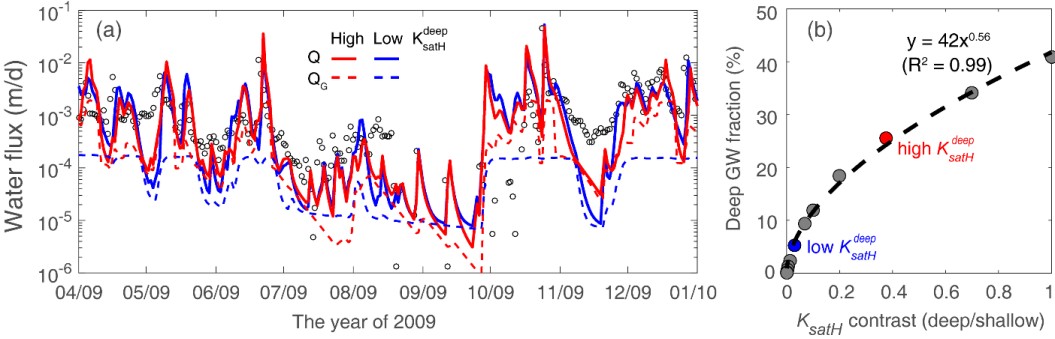


**Figure 6.** (a) Hydraulic conductivity ($K_{satH}$) contrast control on deep groundwater ($Q_G$).
The cases of high (red) and low (blue) $K_{satH}^{deep}$ led to 26% and 5.2% of annual $Q_G$
contribution to discharge (Q), respectively. (b) Deep groundwater fraction as a function of
$K_{satH}$ contrast between the deep and shallow zone. The $K_{satH}$ contrast was limited to 1
in the figure as most watersheds exhibit a smaller $K_{satH}$ in the deep zone than in the
shallow zone. The two red and blue dots correspond to the two cases in left panel.

A series of similar cases were further tested to generate the relationship between

deep groundwater fraction (%) of discharge and $K_{satH}$ contrast (Figure 6b). Results show
that the deep groundwater fraction exponentially increased with the increasing $K_{satH}$
contrast, reaching a limit at when $K_{satH}$ contrast is sufficiently high. The results also
suggest that the maximum deep groundwater contribution to the stream was limited to ~
40% as most watersheds exhibit smaller $K_{satH}^{deep}$ than $K_{satH}^{shallow}$. The fitting function (dashed



line) could be a useful predictor to quantify deep groundwater contribution at headwater
watersheds given measured deep and shallow hydraulic conductivity.

**5.2 Reactive Transport Example 1: Understanding nitrate dynamics using a**
**spatially implicit domain**
This example focuses on nitrate, which is the dominant N form in water and has relatively
abundant measurements from 2008 to 2010 (https://criticalzone.org/shale-
hills/data/datasets/) (Weitzman and Kaye, 2018). Based on their field measurements, the
atmospheric deposition was the dominant N input and export via discharge was only a
small fraction (2.5%) of atmospheric N input. Most deposited N was either lost to the
atmosphere via denitrification or uptaken by trees. The model at Shale Hills watershed
included atmospheric N deposition, soil N leaching, stream export, denitrification, and
plant uptake (Figure 7).

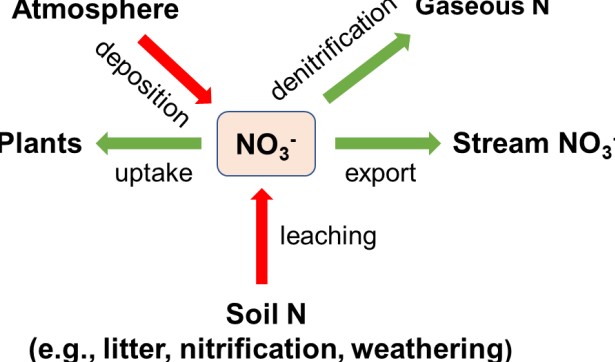


**Figure 7.** Modeled nitrogen processes at Shale Hills. Atmospheric N deposition was the major N
sources (top red arrows); denitrification and plant uptake were the major N loss and sink (green
arrows). Export from discharge removes nitrate but it was a relatively small one.

The soil N leaching process was a lumped reaction that generates $NO_3^-$ source,
including the decomposition of soil organic matter (SOM), nitrification, and rock
weathering. Its rate was assumed to depend on soil temperature and moisture:
$$r_{leach} = kAf(T)f(S_w) \quad (21)$$





Here $r_{leach}$ is the leaching rate (mol/s), $k$ is the rate constant (mol/m²/s), and the surface
area $A$ (m²) is a lumped parameter representing the effective contact area between
substrates and N transforming microbe. It was calculated based on SOM volume fraction
(m³/m³), specific surface area (SSA, m²/g), substrate density (g/cm³), and element
volume (m³).

The denitrification process converts $NO_3^-$ to $N_2$ gas under anaerobic conditions.

This process can be modeled by the Monod rate law with nitrate as the electron acceptor
substrate ($K_{m,NO_3^-} = 45\ uM$ (Regnier and Steefel, 1999;Billen, 1977)) and with inhibition
from $O_2$ (Eq. (22)). Under conditions where $O_2$ concentration is not explicitly modeled (this
work), the $O_2$ inhibitory term can be replaced by a function of soil moisture (Eq. (24)). This
is based on field evidence that denitrification typically occurs when soil moisture is greater
than 0.6 and increases with increasing soil moisture (Brady et al., 2008). Equation (24)
says that under relatively drier conditions ( $S_w < 0.6$ ), there is sufficient $O_2$ that
denitrification does not occur; under wet conditions ($S_w \geq 0.6$), the $O_2$ becomes limiting
such that denitrification can occur.
$$r_{denitrification} = kA\left(\frac{C_{NO_3^-}}{K_{m,NO_3^-} + C_{NO_3^-}}\right)f(O_2)f(T)f(S_w) \quad (22)$$

$$f(O_2) = \frac{K_{I,O_2}}{K_{I,O_2}+C_{O_2}} \quad (23), \text{ when } O_2 \text{ is explicitly modelled}$$

$$f(O_2) = \begin{matrix} 0\ (S_w < 0.6) \\ (S_w - 0.6)*2.5\ (S_w \geq 0.6) \end{matrix} \quad (24), \text{ when } O_2 \text{ is not explicitly modelled}$$


Nitrate uptake by plants is intrinsically complex and not yet completely understood

(Devienne-Barret et al., 2000;Crawford and Glass, 1998;Hachiya and Sakakibara, 2016).
A variety of plant uptake models exists in literature with varying levels of complexity
(Neitsch et al., 2011;Fisher et al., 2010;Cai et al., 2016). These models are mostly based
on plant growth module or supply and demand approach that often requires detailed
phenological and plant attributes such as growth cycle, root age and biomass, nitrate
availability, phosphorous stress, and carbon allocation, in addition to local climate
conditions such as temperature and soil moisture (Neitsch et al., 2011;Porporato et al.,





2003;Dunbabin et al., 2002;Buysse et al., 1996;Fisher et al., 2010). Without all the
detailed information, here we assumed a simple and operational approach to model
nitrate uptake with dependence on $NO_3^-$ concentration, soil temperature and moisture,
and rooting density (Eq. (25), (26)). More detailed, user-tailored plant uptake rate law can
be added if needed.

$$r_{uptake} = k_{uptake} C_{NO_3^-} f(T) f(S_w) f_{root}(d_w) \quad (25)$$

$$f_{root}(d_w) = \exp((-d_w + r)/s) \quad (26)$$

Where $k_{uptake}$ is the nitrate uptake rate (L/s), $f_{root}(d_w)$ is a normalized rooting density
term in the range of 0 to 1 as a function of water depth to the groundwater ($d_w$). The
rooting term (Eq. (26)) was exponentially fitted ($r = 0.0132, s = 0.202$) based on field
measurements of root distribution along depth (Figure S8) (Hasenmueller et al., 2017).
The exponentially declining root function is generally to be the case in forested
watersheds but can be tailored to agricultural watersheds when field data are available.
For microbial soil N leaching and denitrification, reaction rate constant $k$ was
specified (Regnier and Steefel, 1999) and the lumped surface area $A$ (m², = specific
surface area m²/g $\times$ g of mass) was turned to reproduce stream nitrate dynamics and its
C-Q pattern (Table S8). The calibrated effective specific surface area (SSA) were orders
of magnitude lower than the lab measured SSA of natural materials (e.g., SOM, 0.6 ~ 2
m²/g) (Rutherford et al., 1992;Chiou et al., 1990). Such discrepancies between calibrated
effective reactive surface area (i.e., solid-water contact area) and lab measured absolute
surface area are consistent with other observations (Li et al., 2014;Heidari et al., 2017).
The nitrate uptake rate constant $k_{uptake}$ was calibrated to constrain the partitioning of N
transformation flux between denitrification and plant uptake by the ratio of 1:5, a value
estimated from field measurements of gaseous N outputs (3.53 kg-N/ha/yr) and plant N
uptake (18.3 kg-N/ha/yr) (Weitzman and Kaye, 2018). We assumed that the nitrate uptake
rate $k_{uptake}$ of the deep zone (> 2 m in depth) was 1/1000 of that in the shallow zone,
based on the observations that the rooting density exponentially decrease with depth
(Weitzman and Kaye, 2018;Hasenmueller et al., 2017). Groundwater nitrate was





initialized as 0.43 mg/L, the average of measured groundwater concentration during
2009-2010.

**Temporal dynamics.** Three cases were set up to understand and quantify the effects of
different processes in determining the nitrate dynamics (Figure 8). The *transport-only*
case (green line, *tran*) only has N input from precipitation (at 1.4 ± 0.96 mg/L, based on
the 2009 data of NADP PA42 site) and transport without any reactions. It overestimated
stream nitrate data (0.33 ± 0.39 mg/L) throughout the year. The *transport + N reactions*
case (blue line, *tran + N react*) has the denitrification and soil N leaching processes but
not plant uptake. It lowered the nitrate concentration, suggesting their relative minor role
in controlling N. The *transport + N reactions + uptake* case (red line, *tran + N react + upta)*
have all processes. It significantly lowered the nitrate concentration, especially in April-
May and October-December. There were some overestimated short nitrate peaks from
May to July, exhibiting comparable levels of high precipitation nitrate concentration
(Figure 8a). It is noticeable that the three cases (i.e., *transport-only*, *transport + N*
*reactions*, *transport + N reactions + uptake*) almost overlapped (i.e., minimal difference)
at these overestimated short nitrate peaks, suggesting the nitrate-rich precipitation was
not routed into the subsurface where denitrification and plant uptake could occur and
lower the nitrate concentration. In short, hydrology controlled stream nitrate dynamics by
partitioning the nitrate-rich precipitation into surface runoff, shallow lateral flow, and deep
groundwater. Nitrate reactions primarily controlled stream concentration via the
subsurface flow flowpath where the nitrate-rich precipitation undergone significant nitrate
loss and sink, as denitrification and plant uptake only occurred to remove nitrate in the
subsurface but not in surface water.
Comparing the three outfluxes (Figure 8b), nitrate export via discharge (red) was
negligible compared to denitrification (blue) and plant uptake (green). At the annual scale,
stream nitrate export only accounted for 9.5% outfluxes, whereas denitrification and plant
uptake took up 15% and 75% of precipitated $NO_3^-$, respectively. At Shale Hills, rock N
leaching (weathering) is calculated up to 10% of N precipitation.
Although precipitation source occurred primarily from April to August (70% of total
simulation period), larger storm events in October contributed more to the export. Deeper





groundwater had higher nitrate concentration than shallow water, because most plant
uptake occurred in the shallow zone. The nitrate fluxes into the deeper zone however
only contributed 26% of stream nitrate export at the annual scale, due to the relatively
small groundwater contribution (9.5%) to the stream. Denitrification and plant uptake
largely occurred during the wet period, which coincided with the growing season.
Denitrification peaks often showed up after major storm events.

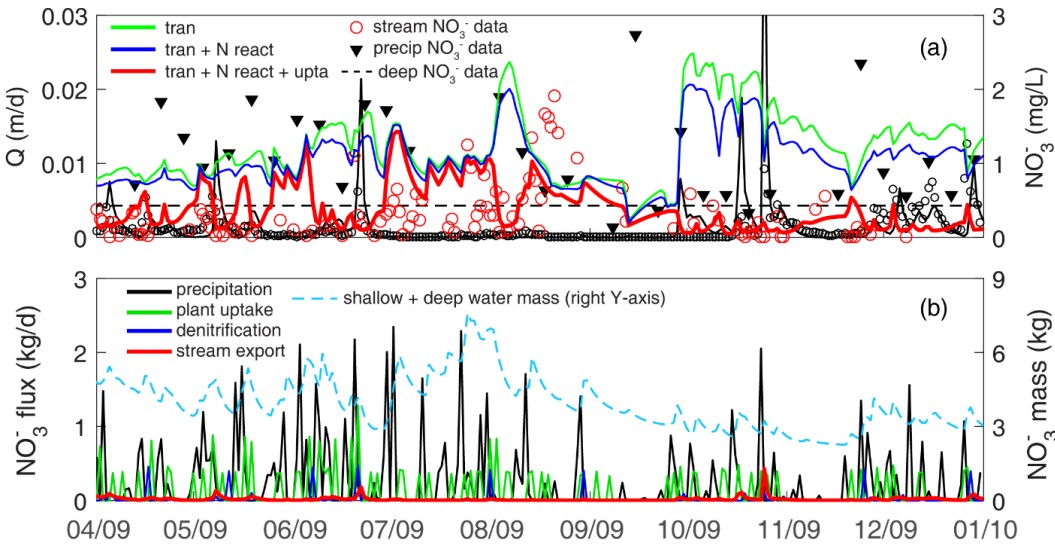


**Figure 8.** Stream nitrate dynamics at Shale Hills in three simulation conditions: *transport-only* (green line, *tran*), *transport + N reaction* (blue line, *tran + N react*), *transport + N reaction + plant uptake* (red line, *tran + N react + upta*), where N reactions include both nitrate leaching and denitrifications (see Figure 8). (a) stream nitrate dynamics; (b) nitrate fluxes and budget. Note the nitrate leaching was ignored in (b) due to its minimal flux as precipitation N deposition was as the dominant input source (Weitzman and Kaye, 2018).

651

**C-Q patterns.** C-Q plots from the three cases showed distinct patterns (Figure 9). Specifically, the *transport-only* (green) and *transport + N reactions* (blue) cases led to chemostatic or slightly flushing patterns while the *transport + N reactions + plant uptake* (red) case showed a dilution pattern similar to field observation. The *transport-only* case showed a slightly flushing pattern because the shallow water had slightly higher nitrate concentration (directly from precipitation without reactions) than deep groundwater. This results in low stream concentrations from deep groundwater at low flow conditions and

high stream concentrations from shallow water with higher nitrate at high flow conditions.
With limited denitrification capacity (Figure 8a), the *transport + N reactions* case was
similar to the *transport-only* case. In comparison, the plant uptake reduced nitrate
concentration in the shallow zone, to an extent lower than the concentration in the deeper
zone, altering the C-Q pattern from primarily chemostatic to dilution (Figure 9).

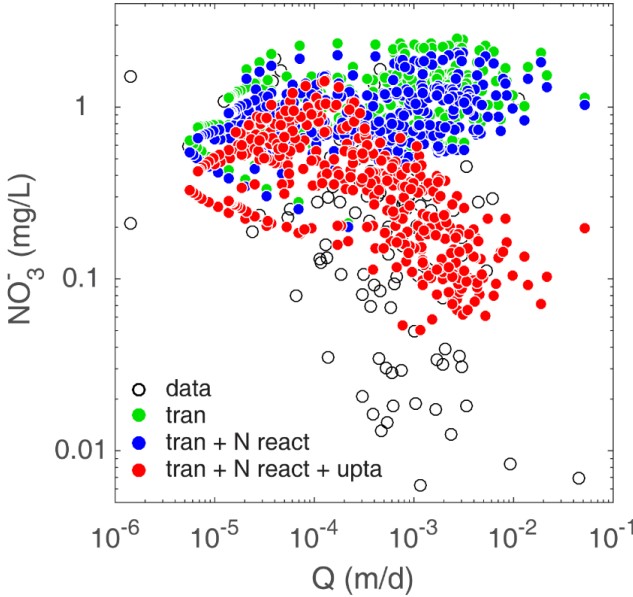


**Figure 9.** Concentration-discharge (C-Q) relationships under three scenarios that involve different processes: *transport-only* (green, *tran*), *transport + N reactions* (blue, *tran + N react*), *transport + N reactions + plant uptake* (red, *tran + N react + upta*).


### 5.3 Reactive Transport Example 2: DOC production and export in a spatially explicit domain

This example showcases the application of BioRT-Flux-PIHM in a spatially explicit
mode. This work has been documented with full details in Wen et al. (2020). Here we only
introduce some key features and capabilities in the spatially explicit mode.
**Model set-up.** In this example, the Shale Hills catchment was discretized into 535
prismatic land elements and 20 stream segments through PIHMgis based on the
topography (Figure 10a). The heterogeneous distributions of soil depth and solid organic





carbon within the domain (Figure 10b-c) were interpolated through ordinary kriging based
on field surveys (Andrews et al., 2011;Lin, 2006). Other soil and mineralogy properties
such as hydraulic conductivity, van Genuchten parameters, and ion exchange capacity
were also spatially distributed following intensive field measurements across the
catchment (Jin and Brantley, 2011;Jin et al., 2010;Shi et al., 2013) (criticalzone.org/shale-
hills/data/).

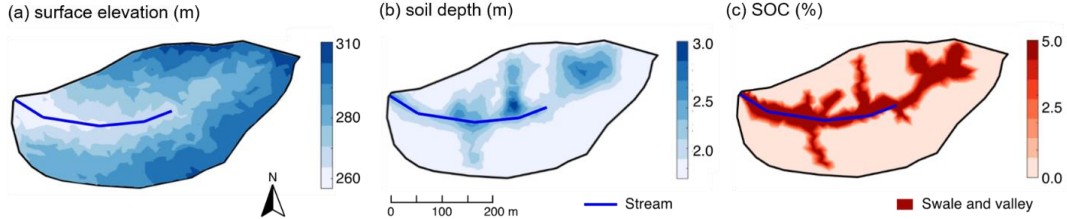


**Figure 10**. Attributes of Shale Hills in the spatially explicit mode: (a) surface elevation, (b) soil
depth, and (c) soil organic carbon (SOC). The surface elevation was generated from lidar
topographic data (criticalzone.org/shale-hills/data); Soil depths and SOC were interpolated using
ordinary kriging based on field surveys (Andrews et al., 2011;Lin, 2006). The SOC distribution in
(c) was further simplified using the high, uniform SOC (5% v/v) in swales and valley soils based
on field survey (Andrews et al., 2011). Swales and valley floor areas were defined based on
surface elevation via field survey and a 10 m resolution digital elevation model (Lin, 2006).

DOC was produced by the decomposition of soil organic carbon (SOC) via the
following reaction:
$$SOC(s) \rightarrow DOC \quad (27)$$
The produced DOC can sorb on soils via the sorption reaction:
$$\equiv X + DOC \leftrightarrow \equiv XDOC \quad (28)$$
where $\equiv X$ and $\equiv XDOC$ represent the functional group without and with sorbed DOC,
respectively (Rasmussen et al., 2018). For DOC production, with abundant SOC and $O_2$
in shallow soils serving as electron donors and acceptors, Eq. (27) can be simplified into
$r_{DOC} = kAf(T)f(S_w)$, where $r_{DOC}$ is the local DOC production rate in individual grids; $k$ is
the kinetic rate constant of net DOC production with a value of $10^{-10}$ mol/m$^2$/s (Zhi et al.,
2019;Wieder et al., 2014); and $A$ is the lumped "surface area" (m$^2$, = $2.5\times10^{-3}$ m$^2$/g × g of
SOC mass) that reflects the effective contact of water with SOC content and biomass





(Chiou et al., 1990;Kaiser and Guggenberger, 2003;Zhi et al., 2019). The temperature
dependence function took the form $f(T) = 2.0^{|T-20|/10}$ while the moisture dependence
function followed $f(S_w) = (S_w)^{1.0}$ (Yan et al., 2018;Hamamoto et al., 2010). In DOC
sorption, equilibrium constant $K_{eq}$ with a value of $10^{0.2}$ ($= \frac{[\equiv XDOC]}{[\equiv X][DOC]}$) represents the
thermodynamic limit of the sorption; The sum of $[\equiv X]$ and $[\equiv XDOC]$ represents the
sorption capacity of the soil with a value ranging from $4.0 \times 10^{-5}$ - $6.0 \times 10^{-5}$ mol/g soil at
Shale Hills (Jin et al., 2010;Li et al., 2017a), depending on the mineralogy.

**Temporal and spatial patterns of DOC production and export.** The model outputs
followed the general trend of stream DOC data (NSE = 0.55 for monthly DOC
concentration; Figure 11a), with high values (~ 15 mg/L) in the dry periods (July-
September). The model enabled the identification of spatial patterns and the hot spots of
reactions. In May when soil water is relatively abundant, valley and swales with deeper
soils (Figure 11b) generally tended to be wetter compared to the hillslope and ridgetop,
and were hydrologically connected to the stream (Figure 11b-c). The distribution of local
DOC production rate $r_{DOC}$ and DOC concentration followed that of SOC (Figure 11c) and
water content (Figure 11b). Low $r_{DOC}$ in relatively dry planar hillslopes and uplands
resulted in low soil water DOC. The average stream DOC (~ 5 mg/L) reflected soil water
DOC in the valley and swales.
In August, the hydrologically-connected zones with high water content shrank to
the vicinity of the stream and river bed. With high temperature in summer, $r_{DOC}$ increased
by 2-fold from May across the whole catchment while still exhibited the highest values in
the SOC-rich regions. Soil water DOC concentration increased by a factor of 2 because
the produced DOC was trapped in low soil moisture areas that were not hydrologically
connected to the stream. In the north side with low water content (Figure 11b), the soil
water DOC (~ 7 mg/L in average) accumulated more than the south side (~ 5 mg/L in
average). The high shallow water DOC (~ 10 mg/L) in the stream vicinity dominated the
stream DOC in August.
In October, precipitation wetted the catchment again. The hydrologically
connected zones expanded beyond swales and valley to the upland hillslopes (Figure



11c). The increase in hydrological connectivity zones favored the mixture of shallow water
DOC sourced from upland hillslopes (low DOC), swales, and valley (high DOC) into
stream rather than only from the stream vicinity with high DOC in the dry August, leading
to a drop in stream DOC.

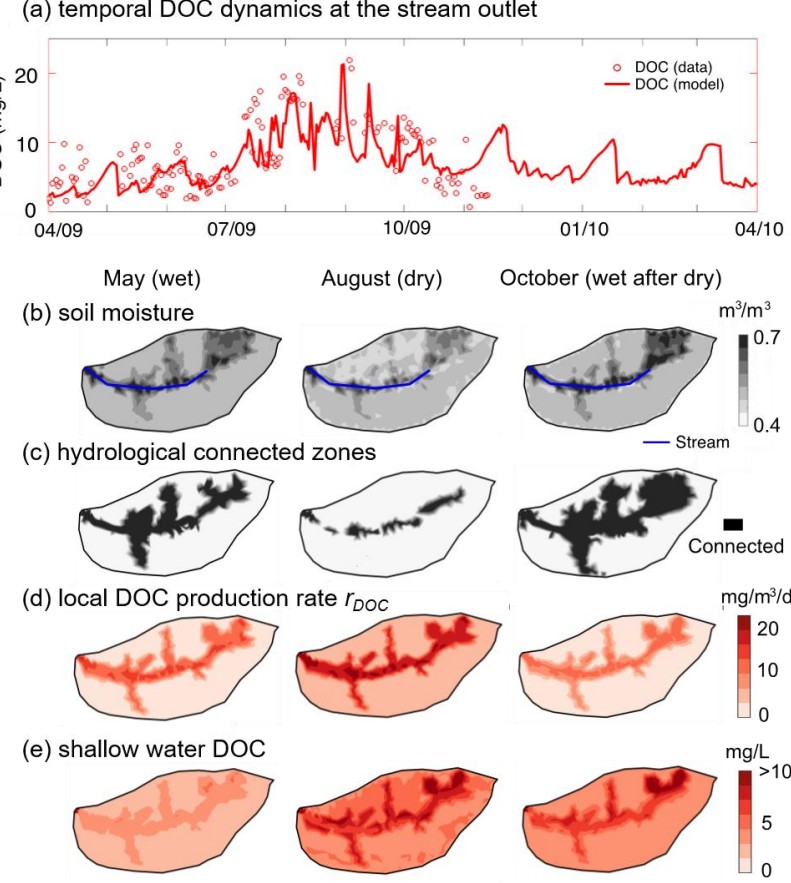


**Figure 11**. (a) Temporal dynamics of stream DOC concentration; spatial profiles of (b) shallow
soil moisture, (c) hydrologically connected zones, (d) local DOC production rates $r_{DOC}$ and (e)
shallow water DOC concentration in May (wet), August (dry), and October (wet after dry) of 2009.
The soil DOC and $r_{DOC}$  were high in swales and valley with relatively high shallow water and SOC
content. August had the highest shallow water DOC concentration compared to May and October,
because most DOC accumulated in zones that are disconnected to the stream.

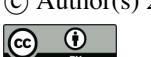

**C-Q patterns.** The DOC C-Q relationship showed a non-typical pattern with flushing first
and transitioning into a dilution pattern, with a general (overall) C-Q slope $b$ = -0.23
(Figure 12). At low discharges (< $1.8 \times 10^{-4}$ m/d) in summer dry period, the stream DOC
mainly came from the organic-rich swales and valley floor zones with high soil water DOC
(Figure 11e). With discharge increasing in wetter period (i.e., spring and fall), the
contribution from planar hillslopes and uplands with lower DOC concentration increased
(Figure 11e), leading to the dilution of stream DOC.

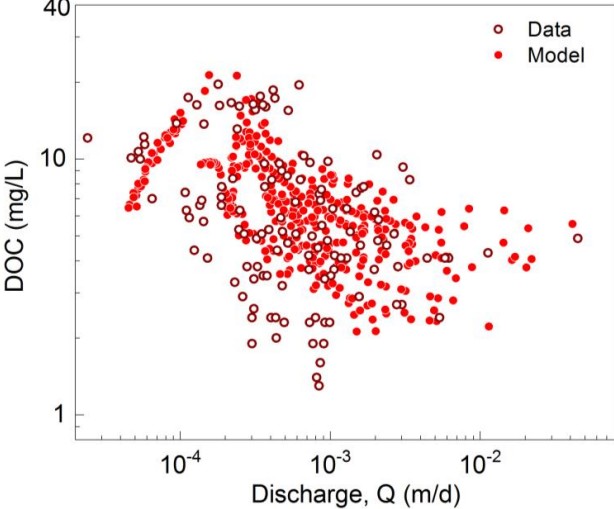


**Figure 12**. Relationships of daily discharge ($Q$) with stream DOC concentration. With the increase
of $Q$, the stream water first shifted from the dominance of groundwater with low DOC at very low
discharge to the predominance of organic-rich soil water from swales and valley at intermediate
discharge. As the discharge increases further, the stream water switches to the dominance of
high flow with lower DOC water from planar hillslopes and uplands, resulting in a dilution C-Q
pattern (modified from Wen et al., (2020)).

Compared to the spatially implicit model, the spatial representation enables the
exploration on the "hot spots" (i.e., swales and riparian zones with high soil water DOC
concentrations in Figure 11e) and their contribution to stream chemistry at different times.
Spatial heterogeneities in watershed properties (e.g., soil types and depth, lithology,
vegetation, biomass, and mineralogy) are omnipresent in natural systems. Yet a general
understanding of the linkage between local catchment features and catchment-scale





dynamics (e.g., stream concentration dynamics and solute export pattern) is still lacking.
Questions such as how the heterogeneous features affect water flow paths, stream water
chemistry, and biogeochemical reaction rates remain largely unanswered. The spatially
explicit model provides a tool to further explore these questions.

## 6. Summary and conclusion

This paper introduces the watershed-scale biogeochemical reactive transport code
BioRT-Flux-PIHM. This code integrates processes of land-surface interactions, surface
hydrology, and multi-component reactive transport. The new development enables the
simulation of 1) biotic reactions including microbe-mediated redox reactions and plant
uptake, and 2) surface water interactions with water from deeper subsurface zones.
BioRT has been verified against the widely used reactive transport code CrunchTope for
soil carbon, nitrogen, and phosphorus processes. The BioRT module has been applied
to understand carbon, nitrogen, and weathering processes in Shale Hills in central
Pennsylvania, Coal Creek in Colorado, and Volcán Chimborazo watershed in Andes in
Ecuador. Here we showcase the modeling capability of surface-groundwater interactions,
transport and reactive transport processes relevant to nitrate and DOC in Shale Hills in
two simulation modes. One is in spatially implicit mode using averaged properties and
another in spatially explicit mode with consideration of spatial heterogeneity. Results
show that the deep groundwater that interacts with stream is primarily controlled by the
hydraulic conductivity contrast between shallow and deep zone. Soil biogeochemical
reactions in shallow soil primarily determines the shallow water chemistry, especially C,
N, and biogenic solutes, under high flow conditions. The spatially implicit method with two
grids can capture the temporal dynamics of average behavior and mass balance; the
spatially explicit running mode can be used to understand the spatial dynamics and to
identify 'hot spots' of reactions.

**Data availability**. Field data (e.g., discharge, stream chemistry) is archived at Shale Hills
data portal: http://criticalzone.org/shale-hills/data/datasets/ or maintained at HydroShare:
https://www.hydroshare.org/group/147.



**Code availability**. The current model release (BioRT-Flux-PIHM v1.0), including
documentation, source code, example data, is available at GitHub repository:
https://github.com/PSUmodeling/BioRT-Flux-PIHM.

**Competing interests.** The authors declare that they have no conflict of interest.

**Author contributions.** LL conceived the model idea and oversaw the model
development. WZ coded the BioRT module, verified the code against the benchmark
reactive transport model CrunchTope, and applied and tested the model at Shale Hills
watershed. YS developed the deep groundwater component and integrated the BioRT-
Flux-PIHM v1.0 into MM-PIHM family. WH, LS, and GHCN tested the code during its
development and contributed their study cases.

**Acknowledgement.** We acknowledge the funding support from the Department of
Energy, Subsurface Biogeochemistry Program DE-SC0020146, National Science
Foundation Hydrological Sciences EAR-1758795. We appreciate data from the
Susquehanna Shale Hills Critical Zone Observatory (SSHCZO) supported by National
Science Foundation Grant EAR – 0725019 (C. Duffy), EAR – 1239285 (S. Brantley), and
EAR – 1331726 (S. Brantley). Data were collected in Penn State's Stone Valley Forest,
which is funded by the Penn State College of Agriculture Sciences, Department of
Ecosystem Science and Management, and managed by the staff of the Forestlands
Management Office.





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
