# Peer review of "BioRT-Flux-PIHM v1.0: a watershed biogeochemical reactive transport model"

_Geoscientific Model Development, 2020_

## Referee Comment (RC1) · Anonymous Referee #1 · 17 Aug 2020

Dear authors, I am rather torn on this paper. On one hand, I strongly believe some of the results of the paper are significant and interesting and without a doubt worth publishing. On the other hand, I am fairly disappointed with the presentation of the manuscript.

Section 3 is poorly written. A lot of terms are not defined, and it seems that there are some inconsistencies. Notations are poor and make the equations hard to read. The structure of the paper is somehow chaotic. Section 5 includes a series of description of new processes, definition of parameters, rates, ... The most important associated issue is the fact that I do not see what motivated the choice of the content of section 3. The hydrology part could be summed up by one or two equations and adequate references (it's basically Darcy and Richards equation on which multiple pages fo-

cus). The reactive transport equation is fairly basic as well. And somehow, these fairly simple concepts are presented in a confusing way and most of the terms are poorly defined, and a lot of equations are redundant. I feel that that section could almost be entirely removed with adequate citations. On the other hand it seems that a significant proportion of the methods are lacking description (and are included within section 5). Another problem to me is that it seems that some key processes (evapotransporation) are barely discussed in the modelling part), despite some modelling results showing a comparison between ET model and data.

This work is obviously built within a collaborative effort but the paper fails to present what was actually done within this research. A good example is how all the complexity from figures 1, 2 and 4 are completely ignored within the model description.

A last important issue regards how the objectives are stated. Reading the title, abstract and introduction and even the equation section really gives the feeling it is a modelling/numerical paper, while the results section goes into important and interesting details about these coupled multi-phyiscs dynamics and predictive applications for large-scale field data. I feel that the first half of the paper does not constitute a proper description of what is coming next.

There are multiple good things about this very relevant work: - model results are interesting - model verification are good - interesting modelling approach - multi-scale multi-physics problem, - ....

and i'm confident this is worthy of a very nice publication. However, this paper, as is, is a poor representation of the work which was performed. In my opinion, an important work of structure has to be done. Considering how rich the results section is, i would maybe suggest to emphasize these very interesting features of the work as the main part of the article. I am sorry if my review sounds very negative. I was somehow upset by the quality disparity between the work itself and the paper writing. I guess it's better in this way.

In the attached files, you can find some detailed line by line comments. These are comments which occur to me "as i am reading", please forgive the tone within that text file. Hopefully, this can help you.

Please also note the supplement to this comment:
https://gmd.copernicus.org/preprints/gmd-2020-157/gmd-2020-157-RC1-supplement.pdf

**Supplement:**

13: hydrolgy processes

32: spatially implicit: what this means is not clear.

- abstract: not clear weather the verification with Crunch is part of the paper

42: microbes ?
42: more general than CO2: gases fluxes?

59-60: unclear sentence

38-64: there are other references which could be worth citing

64: remove "that"
65-69: this is a pretty long sentence.

70: MacQuarrie and Mayer not sure this is the best citation in this sentence.

128: solar radiation?

- Figure 1 : beautiful figure!

- 144: kinetically controlled or at equilibrium

- 145: no eg for kinetically?

- Figure2 : resolution is not great. it may be due to the journal processing of the figures to generate this file. please be careful with resolution.

- eq(1) and (2) : why is the porosity there? in the previous figure caption, it says "h" refers to "head", so i expect meters. in typical Richards equation, this would yield
a specific storage coefficient, for example.    but then in the paragraph, h is the water storage. What does that mean? what units? L of water/ L of porous medium?

- need to be consistent between figure 2 caption and this equation.

- please say where this reaction comes from and define the different terms clearly and properly with units.
- Looking at Li 2019 RiMG, the storage is defined in m of water, but the porosity does not appear.

- eq(2) : sum ij to 1?  this isn't clear.  why is j <= 3?  this notation is unclear. i'd rather have a sum over symbolic "neighbors" than not defining i, j properly and have a "sum up to 1"

- in every equation, "shallow", "lateral", and actual english words should not be in italic. \textrm{} or \mathrm{} in LaTeX, or straight formatting in word.

- 190: this should not be a new paragraph.

- Eq (3) and (4): this sugests that q has units of m3/s. In equations (1) and (2), if h is in the units of meters, q should have units of m/s.
Anyway, there are inconsistencies between the equations. This could be dealt with simply by removing "A" from the equation (and in the text).

- 191: $K\_infil^{shallow}$ : this is an infiltration? from the form of equation (3), it looks like a hydrualic conductivity.

- 192: in the vertical direction? this suggests some kind of anisotropy: is this the case?

- all the notations with sub and superscripts are really not clear. this should be significantly improved for readability. maybe call $q\_rechg$ with a new symbol R, not write "shallow" on every single term, ...

- 198: $K\_eff\ H\_{ij}^{shallow}$ --> very heavhy notation again. why the need of $H\_{ij}$. I guess it could simply be written as $K\_{ij}$ (every other K has $^{shallow}$), i don't see why
a distinction is required for the "effective".

- 199: now the subscript H is used to denote horizontal. I would suggest simply $K\_{ij}$ to say averaging between two adjacent cells.  and if there is some anisotropy, maybe just
distinguish using subscripts z, or x, to avoid confusion between all these symbols.

- eq(7): similar problem. subscript "i" is not defined.
these equations all adopt some form of discretization and are written for a volume "i". maybe some of the notation and clarity issues would be diminished if they
were not written in a discretized fashion, hence removing all the "i" and "j" from the equations.
Because in theory, a subscript "i" should be added to every other variable to be consistent.

- also if there are indeed written in a discretized fashion, somehow volumes and surfaces should appear. they don't. or they're hidden within the variables.
but it's hard to say as almost no units are given and variables are not clearly established.

- eq(8): same as equation (5). Typical Darcy's law, it could be given once, and text could specify where it is used (if needed).

- eq(9) --> please explain the denominator

- line 235: why area fraction instead of volume fraction?

- 238: infiltration hydraulic conductivity --> please change this denomination. hydraulic conductivity of a certain porous media, but not infiltration.

--> section 3.1 shoud be significantly improved. this section does not introduce many new concepts, but somehow is surprisingly unclear, due to very heavy notations, repetitions, ...

Section 3.2
here in equation (14), the discretization factors (Vi and Aij) do appear. Why not in the previous ones?

- (14) the indices around the sum symbol are pretty heavy. maybe just put sum over j(neighbours), and explain in the text than j refers to the neighbours of cell i.  but $N_{i,1}$ to $N_{i,x}$ is confusing
- (14)/256: all units of distance within gradients have been expressed with "d" or "D" and here it's $I_{ij}$. maybe some consistency could be appreciated.

- 250: + gas volume, i guess (or simply remove the parenthesis, i don't think it's important).
- 250: put the m3 in [] somewhere else, like where $V_i$ [m3] is the total volume of gridcell i.

- 252: index of elemets sharing surfaces -->  it's unclear and i don't think it adds anything

- 255: "combined". for consistency i would suggest to stick with "mean" as was done for hydraulic conductivities (is it harmonic?)

- 257: agree that here, q has m3/s, because $A_{ij}$ has been incorporated. Be careful about consistency with previous section and equations where it seemed to me that q was m/s (but again, it was not very clear).

- 261: microbes

- 265: precipitate as carbonate materials?

- 265: transition between sentences can be improved.   "it can oxidize into co2. or it can precipitate. hence it can release co2". it would make more sense, i think, if it was grouped differently:
" oxidize into CO2 which can be released back to the atmosphere or surface water. or it can precipitate".

- 266: chnage CO2 level (not changes, i think).

- Figure3 : resolution is not great again. But nice illustrating figure, it is helpful.

- 276: maybe give chemical form of ammonia. why is No2- within parentheses?

- It is weird that these paragraphs are part of a section called "equations". Maybe another section after line 258 "Biogechoemical processes"?
a first "paragraph" about description, then the "paragraph" describing kinetics (294)
(And maybe section 3 should be modified to governing equations and processes?)

- eq(15)/ line 303 : why the subscript "c5h7o2N" and not simply microorganisms?

- can you give an example of D and A (electron donor and acceptor)? are they linked to the three pools, A and D being the intermediate stages?

- line 307: espectively; "they are the concentrations at which half of the maximum rates are reached for the electron donor and acceptor respectively" i think this can be omitted.
it's part of the reference and is fairly well understood or self-explanatory.

- equation (16) could be summed up in one equation with one symbolic term like (Product)_inhibiteurs $K_{inh}$ / ($K_{inh}$ + $C_{inh}$) and then in the text examples could be given.

- 334-336: with little pore space for air. i'm not a big fan of this terminology. and the overall paragraph is not very clear. I would simplify like
important under conditions where electron donors and acceptors are limited, e.g. anoxic conditions for O2. Under conditions org carb and o2 are abundant, SOM rate loaw ...  eq(19) and the following.

- eq(19) and 340. need to change notation for μ_max. it's not the same unit, it does not bear the same meaning.

- eq(19): which surface area? m2 of what?

- 348: if it is often used, i guess you could include some citations here :)

- 351: accounted for

- eq 20: i would suggest to merge equations 19 and 20.

- eq 20: bm is the declining coefficient? It's simply a characteristic depth. "declining coefficient" is a weird denomination.

361: CVODE? what do the CV stand for?

- you do not specify how you solve set of equations 14 (reactive transport) which are arguably the most complicated to solve.

- 374: at a range of reaction complexity levels?    "a variety of transport conditions" is a bit overselling what has been done in the SI.
In my understanding, there is only one transport condition and 3 investigated reaction networks.

- there are 2 secions "4"?

- 398: i'd rather say something like "negligible, as the associated evolution of hydrological parameters".

- fig 4: you mention a lot of things which were not discussed in the model section. for example, thermal effects (evapotranspiration, solar radation, air temperature) --> how do they impact what has been discussed.
Basically, most of the complexity from figure 1 and figure 4 was absolutely not addressed in the model section. are there some missing references?

- temperature effects: how do they impact thermodyanmic constants, evaporation, ...

- ET: how do you compute that?

- Nash Sutcliffe efficiency? can you explain what that is? or refer to some work regarding that?

- 560: issue within the chemical species

- eq 25-26: some references are needed

- structure of the paper needs to be revised. a huge amount of new variables and informations are given within the applications.

- results: generally nice figures and results. but the leadup to here does not give the results the credit they deserve.

---------- SUPPLEMENTARY
Table S2 : given rates are surface rates, but no surface area are discussed and the kinetic law is missing.

- it would be nice to add the Saturation index of the initial solution and boundary condition solution with respect to apatite.

- figure S2c: looking at h+ concentration on (a) after 1 residence time, it looks like it at 1e-8 mol/L (and going down). in figure s2c, downstream h+ concentration looks higher.
as its probably in equilibrium with apatite, it's surprising that this value is higher than in figure S2(a). can you comment?

- table S3:
        -units of rate constant.
        - what is X_mio?
        -what is CH2O(s) ? s for solid? how do you define their concentrations? what does it represent? i would understand if it represented a surface area but this seems odd.

S12 --> what about flow/transport setup in this simuation? same as for the first? should be indicated

line 128: i think the reference is Figure S6.
- same transport than previously?

table S5: same comments than for table S3

--
S2. what's the exponential factor?

Figure S9: (a) it's really not clear what is observed .. precipitation is the grey on the top, I guess. could you write "precipitation" in gray then?
Please include evapotransporiation (ET) in the caption.
the word "precipitation" in the middle of the frame is surprising.

- Figure S9(c): there is more water in the unsaturated than in the saturated part?

---

## Short Comment (SC1) · 29 Sep 2020

This manuscript presented a recently developed watershed-scale biogeochemical model and its various applications for the two water-table, and DOC and nitrate at Shale Hills watershed. Currently there is an urgent need in the field for such watershed-scale reactive transport codes and the model developed in this manuscript has a wide potential to reach a broad biogeochemistry community. The manuscript is generally well-organized in model materials and has done a decent job validating the model against the benchmark code CrunchTope. I enjoyed reading the most part of the manuscript yet it is lengthy and could be shortened for conciseness. I think the model presented here will be of interesting to many others who are interested in understanding the interaction of land surface, hydrological, and biogeochemical processes. Yet the manuscript

also needs a major revision to reduce its length, re-organize its structure, and make some clarification. Therefore, I am supportive of its publication after the major revision. Some detailed comments are listed below.

Comments: 1) Figure 1 is a very nice conceptual figure. But it is not clear how the ET is calculated, i.e., what form does it has (e.g., evaporation, transpiration, or even snow sublimation). It is also not clear what does the dark green (e.g., microbe-mediated redox reactions) and shallow green (e.g., mineral dissolution and precipitation) means in the legend.

2) Figure 2 is a useful representation for detailed hydrological processes yet I think the author need to make it more explicitly (or highlight) in the figure about the "two water-table" concept, which is a new model development feature for this study. Does the deeper zone have ET process? How does different water flux terms relate to each other (from the water balance perspective)?

3) The hydrological equations are 2.5 pages long and some of them are repetitive with the same set of equation only in different layers. This section can either be shortened or moved into SI.

4) Macropore and its equation are presented yet are not mentioned or discussed in the later part. Is it also a new feature for the model development, if not, consider remove it.

5) Equation 14, not sure how the hydrology model and bioRT coupled together, i.e., need to be more specific about which terms in this equation are from the hydrology model? Are they coupled outside the hydrology model or coupled internally?

6) Consider reduce the section of 3.2

7) Line 359, not sure if this "numeric scheme" is necessary. Not very relevant to other materials.

8) Figure 5, what are the multiple orange ET arrows, from soil, tress, and snow?

9) Font style is not consistent in Table 1.

10) Some references in Table 1 are missing from the reference list. For example, Leila 2020 (preprint DOI maybe?).

11) Line 485, and Line 493, there seems no mentioning about how was the modeled ET calculated. Consider provide an ET equation or a reference.

12) I like the way Figure 6b was presented. This figure makes sense to me and the upper bound of 40% GW contribution is generally consistent with literature. One tiny thing to improve is the small font size of Figure 6a legend, especially the subscript and superscript.

13) 5.2 Reactive Transport Example, this section already has sufficient details about N reaction. This makes me wondering whether the 3.2 section should be shortened for conciseness.

14) 5.3 Reactive Transport Example 2, it seems to me that the model is flexible in model domain setup. In addition to more model inputs (e.g., spatial information), what are other requirements or burdens in using a spatially explicit model.

---

## Referee Comment (RC2) · Anonymous Referee #2 · 4 Nov 2020

It is very useful to have new models so that one can compare different approaches, especially when the models have open source code like this one.

However, with very many existing catchment-scale biogeochemical water quality models, it should be more clearly stated what this model provides that others don't, and how it can shed light on catchment processes that were previously not well understood. Or alternatively how this new software makes the job of the model user easier. For instance, the process descriptions for Phosphorous, DOC and Nitrogen don't look too dissimilar to existing models.

My impression is that this model is toward the complex end of the spectrum when it comes to parametric complexity. I miss an explicit analysis of this. For instance, how many parameters have to be calibrated and can not be sufficiently constrained by

(easily obtainable) measurement or literature values?

Knowledge of this is of vital importance to a user. For instance, if one wants to do a very thorough investigation of the processes in a single catchment one maybe has some time to spend to do a very detailed model setup. However, in some applications one needs to model all the inputs from land into a whole coastline or a large set of lakes. In such an application one often relies on autocalibration and upscaling, and in such applications high parametric complexity can be detrimental. On a similar note, data availability of data that can be used as model drivers can vary between locations. Does this model accomodate for locations with low data availability?

If the stated goal of the model is to be a research model targeted at understanding catchment processes, rather than a model that can also be used as an input source for oceanic models or to be used by government officials to inform policy decisions on a large scale, then this is maybe not as big of a concern. But that could be made more explicit.

Can you argue why the model complexity is justified? Some studies show that simple models can give as good predictions as complex ones while taking much less time and data to deploy. I can see that you have a plot of sensitivity to turning off various nitrate processes, but what about sensitivity to simpler or more complex descriptions of these processes? What is the sensitivity of model results to perturbations in the parameters?

Similarly, what is the sensitivity to subdividing the land into many cells? Is having 100 cells warranted, or can you get just as good predictions just using a couple of cells describing the different land use types? (I understand the argument about identifying hot spots, but it could also be interesting to see if the subdivision has impact on the stream concentration predictions).

What is the calibration process like for the user? Are any autocalibration tools set up for the model?

---

## Author Comment (AC1) · 13 Feb 2021

**Short comments (SCs)**

This manuscript presented a recently developed watershed-scale biogeochemical model and its various applications for the two water-table, and DOC and nitrate at Shale Hills watershed. Currently there is an urgent need in the field for such watershed-scale reactive transport codes and the model developed in this manuscript has a wide potential to reach a broad biogeochemistry community. The manuscript is generally well organized in model materials and has done a decent job validating the model against the benchmark code CrunchTope. I enjoyed reading the most part of the manuscript yet it is lengthy and could be shortened for conciseness. I think the model presented here will be of interesting to many others who are interested in understanding the interaction of land surface, hydrological, and biogeochemical processes. Yet the manuscript also needs a major revision to reduce its length, re-organize its structure, and make some clarification. Therefore, I am supportive of its publication after the major revision.

**Response:** Thank you for your interests in our model. As you can see from our response to other reviewers, we have thoroughly revised the manuscript for conciseness and clarity.

Some detailed comments are listed below.

1) Figure 1 is a very nice conceptual figure. But it is not clear how the ET is calculated, i.e., what form does it has (e.g., evaporation, transpiration, or even snow sublimation). It is also not clear what does the dark green (e.g., microbe-mediated redox reactions) and shallow green (e.g., mineral dissolution and precipitation) means in the legend.

**Response:** The ET is the sum of evaporation, transpiration, and snow sublimation. We have added a sentence about ET method with a reference. An explanation has been added to the legend.

  *Line 243 – 244: "The ET is calculated by the Penman potential evaporation scheme and detailed equations can be found in Shi (2012)."*

  *Line 170 – 171: "For BioRT, the light and dark greens refer to abiotic and biological reactions, respectively."*

2) Figure 2 is a useful representation for detailed hydrological processes yet I think the author need to make it more explicitly (or highlight) in the figure about the "two water-table" concept, which is a new model development feature for this study. Does the deeper zone have ET process? How does different water flux terms relate to each other (from the water balance perspective)?

**Response:** Only the shallow zone has ET process. We have now revised the figure so relationships between different water fluxes are encoded in water equations (Eqn 1, 2 and Eqn S1, S2).

3) The hydrological equations are 2.5 pages long and some of them are repetitive with the same set of equation only in different layers. This section can either be shortened or moved into SI.
**Response:** We agree and have moved much of the content to SI.

4) Macropore and its equation are presented yet are not mentioned or discussed in the later part. Is it also a new feature for the model development, if not, consider remove it.
**Response:** Macropore it is part of the original Flux-PIHM so it is not new. But here we show how macropore affects the hydraulic conductivity of soil matrix. Moved to SI.

5) Equation 14, not sure how the hydrology model and bioRT coupled together, i.e., need to be more specific about which terms in this equation are from the hydrology model? Are they coupled outside the hydrology model or coupled internally?
**Response:** The hydrological module Flux-PIHM provides water storage $S_{w,i}$ and water fluxes $q_{ij}$ for this equation. The solved temperature profile from Flux-PIHM is also provided to the reaction rate. They are coupled internally (see model code details at https://github.com/PSUmodeling/BioRT-Flux-PIHM).
    *Line 437 – 439: "BioRT reads in the model output of water and temperature from Flux-PIHM, and solves the biogeochemical reactive transport equations."*

6) Consider reduce the section of 3.2
**Response:** Reduced substantially.

7) Line 359, not sure if this "numeric scheme" is necessary. Not very relevant to other materials.
**Response:** We have shortened it and added two sentences for solving the reactive transport equation, as other reviewers asked for it.
    *Line 413 – 417: "In BioRT, the transport step is first solved with water by the preconditioned Krylov (iterative) method and the Generalized Minimal Residual Method (Saad and Schultz, 1986). In the following reaction step, all primary species in each finite volume are assembled in a local matrix and then solved iteratively by the Crank-Nicolson and Newton-Raphson method in CVODE (Bao et al., 2017)."*

8) Figure 5, what are the multiple orange ET arrows, from soil, tress, and snow?
**Response:** Figure 5 has been removed to condense the paper

9) Font style is not consistent in Table 1.
**Response:** Thanks for the catch. Changed.

10) Some references in Table 1 are missing from the reference list. For example, Leila 2020 (preprint DOI maybe?).
**Response:** Reference updated. Thanks.

11) Line 485, and Line 493, there seems no mentioning about how was the modeled ET calculated. Consider provide an ET equation or a reference.
**Response:** We have added a sentence about ET method with a reference.
   *Line 243 – 244: "The ET is calculated by the Penman potential evaporation scheme and detailed equations can be found in Shi (2012)."*

12) I like the way Figure 6b was presented. This figure makes sense to me and the upper bound of 40% GW contribution is generally consistent with literature. One tiny thing to improve is the small font size of Figure 6a legend, especially the subscript and superscript.
**Response:** Slightly enlarged in font size. Thanks.

13) 5.2 Reactive Transport Example, this section already has sufficient details about N reaction. This makes me wondering whether the 3.2 section should be shortened for conciseness.
**Response:** The section of 3.2 has been shortened and reorganized as suggested. Much of the reaction rate related materials have been moved to sections in biogeochemical reactions and biological processes.

14) 5.3 Reactive Transport Example 2, it seems to me that the model is flexible in model domain setup. In addition to more model inputs (e.g., spatial information), what are other requirements or burdens in using a spatially explicit model.
**Response:** For the spatially distributed version, it takes longer time to set up the model and run the model. the computational cost can be high. In addition, the model needs more spatial data in order to represent spatial details and capture dynamics at different spatial locations.
   *Line 487 - 488: "It requires much more data and can be computationally expensive but can be used to identify "hot spots" of biogeochemical reactions within a watershed*

---

## Author Comment (AC2) · 13 Feb 2021

Dear authors, I am rather torn on this paper. On one hand, I strongly believe some of the results of the paper are significant and interesting and without a doubt worth publishing. On the other hand, I am fairly disappointed with the presentation of the manuscript.

**Response:** Thanks for comments. We apologize for the confusions in the earlier manuscript. We have thoroughly revised the manuscript for clarity. Specifically,

1) we have defined the scope in the Introduction to focus on the new advance in the model development. References are added for land surface and hydrological process description discussed in previous work.

2) we have shortened the Methodology section, and moved some equations (including the water equation and Monod rate laws) to the Supporting Information. We have also revised all equations for clarity and brevity in notations and their definitions.

3) we have added more model details, including the numerical scheme for solving the reactive transport equation, data needs for setting up spatially lumped and distributed domains, and how to calibrate the model

4) we have also shortened the three model examples such that the new model features stand out

Section 3 is poorly written. A lot of terms are not defined, and it seems that there are some inconsistencies. Notations are poor and make the equations hard to read. The structure of the paper is somehow chaotic. Section 5 includes a series of description of new processes, definition of parameters, rates... The most important associated issue is the fact that I do not see what motivated the choice of the content of section 3. The hydrology part could be summed up by one or two equations and adequate references (it's basically Darcy and Richards equation on which multiple pages focus). The reactive transport equation is fairly basic as well. And somehow, these fairly simple concepts are presented in a confusing way and most of the terms are poorly defined, and a lot of equations are redundant. I feel that that section could almost be entirely removed with adequate citations. On the other hand, it seems that a significant proportion of the methods are lacking description (and are included within section 5). Another problem to me is that it seems that some key processes (evapotranspiration) are barely discussed in

the modelling part), despite some modelling results showing a comparison between ET model and data.

**Response:** Thanks for these comments.

- we have shortened this section significantly by moving the repeated equations to SI and merged some reaction equations as suggested to improve the logic flow.
- we have revised all equations, notations, and their definitions for brevity and clarity.
- we did keep some of the reaction description there so that the readers (and users) know the types of reactions that the model can simulate. BioRT differs from other water quality models that often primarily target a few contaminants (e.g., N, P, metals). The framework of the code is flexible and the users can define their own types of reactions and solutes of interests. We believe it is important for readers to understand that aspect. Please see pages 9 - 16 for revised Section 3.

*Line 131 – 136: "This paper introduces new developments in the BioRT model. The code has been verified against the widely used reactive transport code CrunchTope (Supporting Information, SI). This paper briefly overviews water and energy related processes incorporated in the model. Readers are referred to previous publications for more details of processes such as evapotranspiration (ET), hydrological flow, and abiotic reactions (Shi et al., 2013;Bao et al., 2017;Li et al., 2017a;Qu and Duffy, 2007)."*

*Line 243 – 245: "The ET is calculated by the Penman potential evaporation scheme and detailed equations can be found in Shi (2012). A similar set of water equations for the deep zone are in the SI (Eqn. S1 and S2)."*

*Line 291 - 294: "BioRT differs from general water quality models that often primarily target a few contaminants (e.g., N, P, metals). The framework of the code is flexible and the users can define their reactions and solutes of interests in the input files."*

This work is obviously built within a collaborative effort but the paper fails to present what was actually done within this research. A good example is how all the complexity from figures 1, 2 and 4 are completely ignored within the model description.

**Response:** We have revised the manuscript to focus more on new model developments to suit the scope of the GMD journal, and discuss some but less in the Examples.

A last important issue regards how the objectives are stated. Reading the title, abstract and introduction and even the equation section really gives the feeling it is a modelling/numerical paper, while the results section goes into important and interesting details about these coupled multi-physics dynamics and predictive applications for large-scale field data. I feel that the first half of the paper does not constitute a proper description of what is coming next.

**Response:** Thanks for the comment, and apologies for trying to insert too much materials in one paper. We have decided to focus this manuscript mostly on the model development. As you can see from the revised manuscript, we have reorganized much of the model description and removed a large part of results and discussions, with the remaining figures mostly intended to show the model capability.

There are multiple good things about this very relevant work: - model results are interesting - model verification are good - interesting modelling approach - multi-scale - multi-physics problem, - ....and I'm confident this is worthy of a very nice publication. However, this paper, as is, is a poor representation of the work which was performed. In my opinion, an important work of structure has to be done. Considering how rich the results section is, i would maybe suggest to emphasize these very interesting features of the work as the main part of the article. I am sorry if my review sounds very negative. I was somehow upset by the quality disparity between the work itself and the paper writing. I guess it's better in this way.

**Response:** We agree and we appreciate the comment about directions for change. As stated earlier, we have done a major overhaul of the manuscript following these comments. We have revised the methodology for clarity and brevity but also emphasize more of the new features of the model in the examples.

In the attached files, you can find some detailed line by line comments. These are comments which occur to me "as i am reading", please forgive the tone within that text file. Hopefully, this can help you. Please also note the supplement to this comment: https://gmd.copernicus.org/preprints/gmd-2020-157/gmd-2020-157-RC1-supplement.pdf

**Detailed comments in the attachment:**
13: hydrology processes
**Response:** Changed as suggested.

32: spatially implicit: what this means is not clear.
**Response:** Revised to "spatially lumped".

- abstract: not clear whether the verification with Crunch is part of the paper
**Response:** We have added clarifications:
>     Lines 131 – 133: "The code has been verified against the widely used reactive transport code CrunchTope (Supporting Information, SI)."

*Line 419 – 420: "The BioRT module had been verified against CrunchTope under different transport and reaction conditions (Figures S1 – S7 in SI)."*

42: microbes?
**Response:** Changed.

42: more general than CO2: gases fluxes?
**Response:** Changed.

59-60: unclear sentence
**Response:** We have rephrased the long sentence.
*Line 65 – 68: "The lack of understanding of mechanisms governing hydrological and biogeochemical interactions presents major roadblocks for forecasting water quality, including water issues such as eutrophication that persist worldwide".*

38-64: there are other references which could be worth citing
**Response:** We have added multiple references in the Introduction.

64: remove "that"
**Response:** Removed.

65-69: this is a pretty long sentence.
**Response:** This long sentence has been rephrased to the following:
*Line 73 – 78: "Hydrologic models focus on solving for water storage and fluxes at the watershed scale and beyond (Fatichi et al., 2016). Reactive transport models (RTMs) have traditionally centered on transport and multi-component biogeochemical reactions typically in groundwater systems, which often have limited interactions with climate and other surficial watershed processes (Steefel et al., 2015;Li et al., 2017b;Mayer et al., 2002)."*

70: MacQuarrie and Mayer not sure this is the best citation in this sentence.
**Response:** Citation removed.

128: solar radiation?
**Response:** Yes, the model includes solar radiation processes as suggested in Figure 1.

- Figure 1: beautiful figure!
**Response:** Thanks.

- 144: kinetically controlled or at equilibrium

**Response:** Changed to "kinetically controlled" or "equilibrium-controlled".

- 145: no eg for kinetically?

**Response:** added an example of "microbial redox reaction".

*Line 177 – 179: "The reactions can be kinetically controlled (e.g., microbial redox reaction) or equilibrium-controlled (e.g., ion exchange, surface complexation (sorption), and aqueous complexation)."*

- Figure 2: resolution is not great. it may be due to the journal processing of the figures to generate this file. please be careful with resolution.

**Response:** We have re-generated the figure a 300 dpi resolution.

- eq (1) and (2): why is the porosity there? in the previous figure caption, it says "h" refers to "head", so i expect meters. in typical Richards equation, this would yield a specific storage coefficient, for example. but then in the paragraph, h is the water storage. What does that mean? what units? L of water/ L of porous medium?

**Response:** The "h" refers to water storage [m] and the fluxes in Eqn 1 and 2 are area-normalized fluxes in the unit of [m/s]. The storage h here is essentially the height of soil column with fully saturated water, not the height of the 100% water column. That is why the porosity term $\theta_i^{sl}$ [$m^3$ pore space/$m^3$ total volume] is there. We added the following to clarify:

*Line 232 – 236: "Note that the storages $h$ here are essentially the height of soil column with equivalent saturated water, not the height of the pure water (100% volume) column. That is why porosity is in the equation. For saturation zones, this height is needed to quantify the depths of water tables and determines the direction of water flow between neighboring grids."*

- need to be consistent between figure 2 caption and this equation.

**Response:** Revised both Figure 2 and Eqn 1 and 2 to be consistent. Thanks.

- please say where this reaction comes from and define the different terms clearly and properly with units. Looking at Li 2019 RiMG, the storage is defined in m of water, but the porosity does not appear.

**Response:** As discussed earlier, we have added some clarification about this. The Storage (S) equation (5) and (6) in Li (2019) is actually the same as ($\theta \times h$) here. The equations in Li 2019 were meant to represent the general water balance of the system so it is not necessary to represent the storage as ($\theta \times h$). In the specific code BioRT, we do

need to calculate the height of saturated water table in porous media to infer the direction of flow among neighboring grids so this soil water height calculation is needed.

- eq (2): sum ij to 1? this isn't clear. why is j <= 3? this notation is unclear. I'd rather have a sum over symbolic "neighbors" than not defining i, j properly and have a "sum up to 1"
- in every equation, "shallow", "lateral", and actual English words should not be in italic. \textrm or \mathrm in LaTeX, or straight formatting in word.

**Response:** We have changed the notation and added an explanation for it. We used a straight formatting in the built-in Equation in Word to be consistent.

*Line 241 – 243: "$N_{ij}$ ($\leq 3$) is the number of neighbor elements $j$. For a prismatic element $i$, a boundary cell could have one or two neighbors; a non-boundary cell has three neighbors."*

- 190: this should not be a new paragraph.

**Response:** Changed.

- Eq (3) and (4): this suggests that q has units of m3/s. In equations (1) and (2), if h is in the units of meters, q should have units of m/s. Anyway, there are inconsistencies between the equations. This could be dealt with simply by removing "A" from the equation (and in the text).

**Response:** Apologies for the mistake. Area A has been removed from Eqn. 3 and 4 to be consistent in flux (m/s). Thanks for the catch.

- 191: K_infil ^shallow: this is an infiltration? from the form of equation (3), it looks like a hydraulic conductivity.

**Response:** This is the hydraulic conductivity of the infiltration layer, which is the top 0.1 m of the subsurface and is considered to have different conductivity from the rest of subsurface. We modified the text.

*Line 253 – 255: "$K_{i,inf}$ [m/s] is the hydraulic conductivity of the infiltration layer, the top 0.1 m of the subsurface and is considered to have different conductivity from the rest of subsurface;"*

- 192: in the vertical direction? this suggests some kind of anisotropy: is this the case?

**Response:** Correct, hydraulic conductivity often differs in horizontal ($K_H$) and vertical ($K_V$) direction. The users have the option to set up the different conductivity in different directions.

- all the notations with sub and superscripts are really not clear. this should be significantly improved for readability. maybe call q_rechg with a new symbol R, not write "shallow" on every single term, ...

**Response:** Thanks for the suggestion. We have changed the subscript and superscript of the notations. We would like to keep the q to be consistent with all other flow but have revised the definitions for clarity. In addition, "R" and "r" have been used for total and specific reaction rates (see Eqn. 6, 7). Please sees Eqn. 1 – 5 and relevant description for water equations.

- 198: K_eff H_ij ^shallow --> very heavhy notation again. why the need of H_ij. I guess it could simply be written as K_ij (every other K has ^shallow), i don't see why a distinction is required for the "effective".

**Response:** We have followed the suggestions and revised all notations and removed "eff" in K.

Line 255 – 256: "$K_{i,V}^{sl}$ [m/s] is the hydraulic conductivity in the vertical direction (i.e., weighted average of macropore $K_{i,macV}$ and soil matrix $K_{i,satV}$, Eqn. S7); "

- 199: now the subscript H is used to denote horizontal. I would suggest simply K_ij to say averaging between two adjacent cells. and if there is some anisotropy, maybe just distinguish using subscripts z, or x, to avoid confusion between all these symbols.

**Response:** Thanks for the suggestion. We have removed "H" and simplified as suggested. See our revised Eqn.2 and 5 and relevant descriptions.

Eqn. 2:

$$\theta_i^{sl} \frac{dh_{i,s}^{sl}}{dt} = q_{i,u2s}^{sl} - q_{i,rechg} - e_{i,s}^{sl} + \sum_1^{N_{ij}} q_{ij}^{sl} \quad (2)$$

Eqn. 5:

$$q_{ij}^{sl} = K_{ij}^{sl} \frac{H_{i,s}^{sl} - H_{j,s}^{sl}}{d_{ij}} \quad (5)$$

- eq (7): similar problem. subscript "i" is not defined. these equations all adopt some form of discretization and are written for a volume "i". maybe some of the notation and clarity issues would be diminished if they were not written in a discretized fashion, hence removing all the "i" and "j" from the equations. Because in theory, a subscript "i" should be added to every other variable to be consistent.

**Response:** We have added subscript "i" to all equations. Thanks.

- also if there are indeed written in a discretized fashion, somehow volumes and surfaces should appear. they don't. or they're hidden within the variables. but it's hard to say as almost no units are given and variables are not clearly established.

**Response:** The governing equations are in the ordinary differential form (in terms of time). In the water equation, all fluxes with volume units have been normalized by surface/area to be in the unit of [m/s], to be consistent with what is typically done in hydrologic literature. Volume is implicit there if we time the q terms with corresponding area. The volume and surfaces are in the reactive transport equation (Eqn. 6).

- eq (8): same as equation (5). Typical Darcy's law, it could be given once, and text could specify where it is used (if needed).

**Response:** We agree. This has been moved to Supporting Information.

- eq (9) --> please explain the denominator

**Response:** Explanation added in the SI.

> SI, *Line 24 − 25:* "$0.5\,[H_{i,s}^{sl} + (d_i^{dp} − H_{i,s}^{dp})]$ *is the distance between the center of shallow saturated zone and the center of the deep unsaturated zone (i.e.,* $d_i^{dp} − H_{i,s}^{dp}$).*"*

- line 235: why area fraction instead of volume fraction?

**Response:** Because the hydraulic conductivity depends on how much area in the direction vertical to the flow is macropore and how much is soil or rock matrix. Added in SI:

> SI, *Line 51 − 53: "The conductivity depends on area fraction instead of volume fraction, because the hydraulic conductivity depends on how much area in the direction vertical to the flow is macropore and how much is soil or rock matrix."*

- 238: infiltration hydraulic conductivity --> please change this denomination. hydraulic conductivity of a certain porous media, but not infiltration.

**Response:** Changed.

- section 3.1 should be significantly improved. this section does not introduce many new concepts, but somehow is surprisingly unclear, due to very heavy notations, repetitions, ...

**Response:** We have shortened the section. Water equations in the deep zone have been moved to SI. All equations and notations (e.g., subscript and superscript) have been revised for clarity. Thanks.

- section 3.2 here in equation (14), the discretization factors (Vi and Aij) do appear. Why not in the previous ones?

**Response:** The water equations (e.g., Eqn. 1 – 5) were all written in fluxes terms [m/s] because if we use volume / time, the equations will need to involve two types of areas, one is the land area in the horizontal direction, the other is the area between neighboring prismatic grids, which can become confusing. If the reviewer thinks this is essential, we can change the equations to forms that involve volume in the next revision.

For reactive transport equations (Eqn. 6), since they are mass balance equations, it is much easier to have volume, because all concentrations are related to volume. Mathematically we can normalize everything by land area but then the units will become confusing. So, for now we are keeping the current forms.

- (14) the indices around the sum symbol are pretty heavy. maybe just put sum over j (neighbours), and explain in the text than j refers to the neighbors of cell i. but N_i,1 to N_i,x is confusing.
**Response:** We have revised the equation to the following:

*Line 273:* "$V_i \frac{d(S_{w,i}\theta_i C_{m,i})}{dt} = \sum_1^{N_{ij}} \left( A_{ij} D_{ij} \frac{C_{m,j}-C_{m,i}}{d_{ij}} - q_{ij} A_{ij} C_{m,j} \right) + R_{m,i}, \; m = 1, \dots, nm$ (6)"

*Line 276 – 280:* "$N_{ij}$ *is the number of fluxes from neighbor element $j$ for element $i$, $N_{ij}$ is 2 for the unsaturated zone (infiltration, recharge) with only vertical flows and 5 for saturated zone with flux from (or to) the unsaturated zone, from (or to) the deeper zone, and fluxes between $i$ and three neighbor elements $j$ in lateral flow directions for non-boundary grids;*"

- (14)/256: all units of distance within gradients have been expressed with "d" or "D" and here it's I_ij. maybe some consistency could be appreciated.
**Response:** We have changed all distance notation to "d".

- 250: + gas volume, i guess (or simply remove the parenthesis, i don't think it's important).
**Response:** Removed.

- 250: put the m$^3$ in [] somewhere else, like where Vi [m$^3$] is the total volume of grid cell i.
**Response:** Changed as suggested.

- 252: index of elements sharing surfaces --> it's unclear and i don't think it adds anything
**Response:** We have revised the indexes:

*Line 276 – 280:* "$N_{ij}$ *is the number of fluxes from neighbor element $j$ for element $i$, $N_{ij}$ is 2 for the unsaturated zone (infiltration, recharge) with only vertical flows and 5 for saturated zone with flux from (or to) the unsaturated zone, from (or to) the deeper zone,*

*and fluxes between $i$ and three neighbor elements $j$ in lateral flow directions for non-boundary grids;"*

- 255: "combined". for consistency i would suggest to stick with "mean" as was done for hydraulic conductivities (is it harmonic?)
**Response:** We have changed the sentence.
  *Line 281 – 283: "$D_{ij}$ [$m^2$/s] is the hydrodynamic dispersion coefficient (i.e., sum of mechanical dispersion and effective diffusion coefficient) normal to the shared surface $A_{ij}$;"*

- 257: agree that here, q has $m^3$/s, because Aij has been incorporated. Be careful about consistency with previous section and equations where it seemed to me that q was m/s (but again, it was not very clear).
**Response:** Area $A_{ij}$ has been added to the $q_{ij}$ term (Eqn. 6). All fluxes (e.g., Eqn. 1 – 5) are now in the unit of [m/s] for consistency.
  *Line 273: "$V_i \frac{d(S_{w,i}\theta_i C_{m,i})}{dt} = \sum_1^{N_{ij}} \left( A_{ij}D_{ij} \frac{C_{m,j}-C_{m,i}}{d_{ij}} - q_{ij}A_{ij}C_{m,j} \right) + R_{m,i}, \ m =$*
*$1,\dots,nm \quad (6)"$*

- 261: microbes
**Response:** Changed.

- 265: precipitate as carbonate materials?
**Response:** We have rephrased the sentence.
  *Line 301 – 302: "With coexisting cations (e.g., Ca, Mg), DIC can often precipitate and become carbonate minerals (e.g., $CaCO_3$)."*

- 265: transition between sentences can be improved. "it can oxidize into $CO_2$. or it can precipitate. hence it can release $CO_2$". It would make more sense, i think, if it was grouped differently: "oxidize into CO2 which can be released back to the atmosphere or surface water. or it can precipitate".
**Response:** We have re-organized the sentence.
  *Line 297 – 301: "SOM can be decomposed partially into organic molecules that dissolve in water (Wieder et al., 2015), i.e., DOC, or it can be oxidized completely into CO2 that is released back to the atmosphere as a gas (Davidson, 2006) or surface water in the form of dissolved inorganic carbon (DIC)."*

- 266: chnage CO2 level (not changes, i think).
**Response:** This part has been deleted due to rephrase.

- Figure 3 resolution is not great again. But nice illustrating figure, it is helpful.
**Response:** Thanks. We have made sure all the upload figures are in 300 dpi resolution.

- 276: maybe give chemical form of ammonia. why is $NO_2^-$ within parentheses?
**Response:** Changed.
  *Line 309 – 310: "OM decomposition releases organic nitrogen (R-NH₂), which can further react to become $NH_4^+$, and other nitrogen forms ($N_2$, $N_2O$, $NO$, $NO_2^-$, $NO_2$) (Figure 3)."*

- It is weird that these paragraphs are part of a section called "equations". Maybe another section after line 258 "Biogechoemical processes"? a first "paragraph" about description, then the "paragraph" describing kinetics (294) (And maybe section 3 should be modified to governing equations and processes?)
**Response:** Changed as suggested. Now this section is named as "*3.3 Biogeochemical processes and reaction kinetics*".

- eq (15)/ line 303: why the subscript "C5H7O2N" and not simply microorganisms?
**Response:** Good point. we have changed all to "B$_{micro}$" to represent biomass of microorganisms in relevant equations.

- can you give an example of D and A (electron donor and acceptor)? are they linked to the three pools, A and D being the intermediate stages?
**Response:** Electron donors are typically dissolved organic carbon (DOC) that can be oxidized and become inorganic carbon, and electron acceptors are chemicals that can be reduced. In the denitrification example, the electron donor is DOC and the electron acceptor is nitrate. They are the reactants.
  *SI, Line 74 – 80: "The denitrification rates can be represented by:*

$$r_{NO_3^-} = \mu_{max,NO_3^-} B_{micro} \frac{C_D}{K_{m,D} + C_D} \frac{C_{NO_3^-}}{K_{m,A} + C_{NO_3^-}} \frac{K_{I,O_2}}{K_{I,O_2} + C_{O_2}} \quad (S11)$$

*Here $C_D$ is the concentration of electron donor such as organic matter or carbon (Di Capua et al., 2019); $C_{NO_3^-}$ is the concentration of electron acceptor nitrate; $K_{I,O_2}$ is the inhibition coefficient of $O_2$, or the $O_2$ concentration at which it inhibits denitrification."*

- line 307: espectively; "they are the concentrations at which half of the maximum rates are reached for the electron donor and acceptor respectively" i think this can be omitted. it's part of the reference and is fairly well understood or self-explanatory.
**Response:** Omitted as suggested.

- equation (16) could be summed up in one equation with one symbolic term like (Product)_inhibiteurs K_inh / (K_inh + C_inh) and then in the text examples could be given.

**Response:** Thanks for the suggestion. Changed to the following:

$$r = \mu_{max} B_{micro} \frac{C_D}{K_{m,D} + C_D} \frac{C_A}{K_{m,A} + C_A} \prod \frac{K_{I,H}}{K_{I,H} + C_H} \quad (S10)$$

- 334-336: with little pore space for air. I'm not a big fan of this terminology. and the overall paragraph is not very clear. I would simplify like important under conditions where electron donors and acceptors are limited, e.g. anoxic conditions for O2. Under conditions org carb and o2 are abundant, SOM rate loaw ... eq(19) and the following.

**Response:** We have re-organized the paragraph. The two conditions have been separately as suggested.

*Line 327 – 330: "For example, in shallow oxic soils where organic carbon and $O_2$ are often abundant, the rate law for carbon decomposition can be simplified to the following form assuming microorganism concentrations are relatively constant.*

$$r = kAf(T)f(S_w)f(Z_w) \quad (7)$$

*SI, Line 56 – 59: "Under conditions where electron donors and acceptors are limited, especially anoxic conditions, the kinetics of microbe-mediated reactions can be described by the general dual Monod rate law (Monod, 1949):*

$$r = \mu_{max} B_{micro} \frac{C_D}{K_{m,D} + C_D} \frac{C_A}{K_{m,A} + C_A} \quad (S9)$$

- eq (19) and 340. need to change notation for $\mu$_max. it's not the same unit, it does not bear the same meaning.

**Response:** Changed to rate constant $k$ [mol/m²/s] in *Line 330 – 331.*

- eq (19): which surface area? m2 of what?

**Response:** A lumped parameter to represents SOM content and biomass abundance. It could be estimated based on experimental surface area measurement of SOM or biomass.

*Line 331 – 333: "the surface area A [m²] is a lumped parameter that quantitatively represents SOM content and biomass abundance,"*

- 348: if it is often used, i guess you could include some citations here :)

**Response:** References added.

*Line 351 – 353: "A typical n value is 2 (Yan et al., 2018) with a range between 1.2 and 3.0 (Hamamoto et al., 2010), depending on soil structure and texture."*

- 351: accounted for
**Response:** Changed.

- eq 20: i would suggest to merge equations 19 and 20.
**Response:** Merged as suggested, now as Eqn. 7.
*Line 330:* $r = kAf(T)f(S_w)f(Z_w)$   (7)

- eq 20: bm is the declining coefficient? It's simply a characteristic depth. "declining coefficient" is a weird denomination.
**Response:** changed to "depth coefficient". Thanks.

361: CVODE? what do the CV stand for?
**Response:** CVODE is short for C-language Variable-coefficients ODE solver. So "CV" stands for C-language Variable-coefficients.
*Line 410 – 411: "solved in CVODE (short for C-language Variable-coefficients ODE solver, https://computing.llnl.gov/projects/sundials/cvode),"*

- you do not specify how you solve set of equations 14 (reactive transport) which are arguably the most complicated to solve.
**Response:** A few sentences added for the reactive transport equation.
*Line 413 – 417: "In BioRT, the transport step is first solved with water by the preconditioned Krylov (iterative) method and the Generalized Minimal Residual Method (Saad and Schultz, 1986). In the following reaction step, all primary species in each finite volume are assembled in a local matrix and then solved iteratively by the Crank-Nicolson and Newton-Raphson method in CVODE (Bao et al., 2017)."*

- 374: at a range of reaction complexity levels? "a variety of transport conditions" is a bit overselling what has been done in the SI. In my understanding, there is only one transport condition and 3 investigated reaction networks.
**Response:** There are two different transport conditions (Table S1) plus three reaction scenarios. We have rephrased the sentence as following:
*Line 419 – 420: "The BioRT module had been verified against CrunchTope under different transport and reaction conditions (Figures S1 – S7 in SI)."*

- there are 2 sections "4"?
**Response:** Thanks for the catch. Corrected.

- 398: I'd rather say something like "negligible, as the associated evolution of hydrological parameters".

**Response:** We have rephrased the sentence.

*Line 439 – 442: "At the time scale of months to years that are typical for BioRT-Flux-PIHM simulations, alterations in solid phase properties, including, porosity, permeability, and reactive surface area, are considered negligible such that hydrological parameters remain constant with time."*

- fig 4: you mention a lot of things which were not discussed in the model section. for example, thermal effects (evapotranspiration, solar radiation, air temperature) --> how do they impact what has been discussed. Basically, most of the complexity from figure 1 and figure 4 was absolutely not addressed in the model section. are there some missing references?

**Response:** Sorry for the confusion. This paper is to introduce BioRT-Flux-PIHM focusing on recent new developments. So some of the complexity in Figure 1 or Figure 4 regarding land surface, hydrological, and reactive transport processes have been described and explored in previous papers (Bao et al., 2017;Shi et al., 2013). For example, ET is a core process in the land-surface and hydrological model (Flux-PIHM) and its interactions with soil temperature, sensible and latent heat fluxes (surface energy balance) have been explored in Shi et al. (2013). As mentioned earlier, we also revised the intro to better define the scope.

*Line 113 – 136: "This paper introduces new developments in the BioRT model. The code has been verified against the widely used reactive transport code CrunchTope (Supporting Information, SI). This paper briefly overviews water and energy related processes incorporated in the model. Readers are referred to previous publications for more details of processes such as evapotranspiration (ET), hydrological flow, and abiotic reactions (Shi et al., 2013;Bao et al., 2017;Li et al., 2017a;Qu and Duffy, 2007)."*

We also added the following in the caption of Figure 5 to clarify the scope,

*Line 474 – 477: "This paper focuses on the BioRT component. The land-surface, hydrological processes, and abiotic reactive transport processes have been described in previous papers (Bao et al., 2017;Shi et al., 2013). Discussions on how air temperature and ET influence stream chemistry can be found in Li (2019)."*

- temperature effects: how do they impact thermodynamic constants, evaporation, - ET: how do you compute that?

**Response:** The ET process is coded in the Flux-PIHM and calculated by the Penman potential evaporation scheme. We have added one sentence in the text with reference.

*Line 243 – 244: "The ET is calculated by the Penman potential evaporation scheme and detailed equations can be found in Shi (2012)."*

We also added more detailed description (*Line 339 – 368*) and Figure 4 of rate dependence on T and soil moisture in the manuscript.

[Figure]

**Figure 4**. *(a) Function form of soil temperature dependence and (b, c) soil moisture dependence for reaction rates. The temperature factor $f(T)$ is a function of the $Q_{10}$ (defined by users) and soil temperature. The soil moisture factor $f(S_w)$ is a function of two user-defined parameters $S_{w,c}$ and $n$ and soil water saturation $S_w$. The soil moisture function can represent three types of behaviors: the threshold behavior (b, $0 < S_{w,c} < 1$), increase behavior (red in (c), $S_{w,c} = 1$), and decrease behavior (blue in (c), $S_{w,c} = 0$). Values of $n$ = 1 leads to a linear threshold dependence of $S_w$ while $n < 1$ and $n > 1$ lead to concave and convex dependences, respectively.*

- Nash Sutcliffe efficiency? can you explain what that is? or refer to some work regarding that?
**Response:** Deleted as we have shortened the three examples.

- 560: issue within the chemical species
**Response:** Corrected as the following:
*Line 579 – 585: "Here this process was modeled by the Monod rate law with DOC as the electron donor (Di Capua et al., 2019), $NO_3^-$ as the electron acceptor, and with an inhibition term $f(O_2)$ (Eqn. S13). The reaction rate: $r_{denitrification} = kA \frac{C_{DOC}}{K_{m,DOC}+C_{DOC}} \frac{C_{NO_3^-}}{K_{m,NO_3^-}+C_{NO_3^-}} f(O_2)f(T)f(S_w)$, where $k$ = $10^{-10}$ [mol/m²/s] is the denitrification rate constant (Regnier and Steefel, 1999), half-saturation constants $K_{m,DOC} = 15 [uM]$ and $K_{m,NO_3^-} = 45 [uM]$ (Regnier and Steefel, 1999;Billen, 1977)."*

- eq 25-26: some references are needed
**Response:** References added. Thanks.

- structure of the paper needs to be revised. a huge amount of new variables and information are given within the applications.

**Response:** This Example 2 has been rearranged, with most of the reaction rate related description in method in main text or SI.

- results: generally nice figures and results. but the leadup to here does not give the results the credit they deserve.

**Response:** Now we have defined the scope of this work in the Introduction and shortened the methodology part (especially the water equation part moved to SI) to focus on new developments in the BioRT model. We have organized the results to focus more on the modeling capabilities. We appreciate the reviewer's careful reading and detailed comments.

---------- SUPPLEMENTARY

Table S2: given rates are surface rates, but no surface area are discussed and the kinetic law is missing.

**Response:** We have added the TST rate law with the surface area terms.

*SI, Line 141 – 146: "The apatite dissolution rate is based on the Transition State Theory (TST) (Helgeson et al., 1984), as described by the following:*

$$r_{TST} = Ak(1 - \frac{IAP}{K_{eq}}) \quad (S15)$$

*Where $r_{TST}$ [mol/s] is the mineral dissolution rate, $A$ [m²] is the mineral surface area, $k$ [mol/m²/s] is the rate constant, $IAP$ is the ion activity product, $K_{eq}$ is the equilibrium constant.*

- it would be nice to add the Saturation index of the initial solution and boundary condition solution with respect to apatite.

**Response:** added.

*SI, Line 146 – 147: "The initial and boundary saturation index (i.e., $log_{10}(IAP/K_{eq})$) is -28.9 and -20.9, respectively."*

- figure S2c: looking at h+ concentration on (a) after 1 residence time, it looks like it at 1e-8 mol/L (and going down). in figure s2c, downstream h+ concentration looks higher. as its probably in equilibrium with apatite, it's surprising that this value is higher than in figure S2(a). can you comment?

**Response:** thanks for the catch. We have corrected the "1 residence time" to be 0.5 residence time.

*SI, Line 161: "… spatial pattern plotted at 0.5 residence time (c, d)."*
*SI, Line 221: "… at 0.5 residence time (b)."*
*SI, Line 257: "… at 0.5 residence time (b)."*

- table S3: - units of rate constant. - what is X_mio? - what is CH2O(s)? s for solid? how do you define their concentrations? what does it represent? I would understand if it represented a surface area but this seems odd.

**Response:** We have updated the reaction rate expression in Table S3 and S5 for consistency with microbial equation (Eqn. S9 – S11). The rate and microbial terms are described in *Line 52 – 89*.

S12 --> what about flow/transport setup in this simulation? same as for the first? should be indicated

**Response:** transport conditions have added.

*SI, Line 188 – 189: "The carbon case was tested under the full transport condition with advection, diffusion, and dispersion."*

*SI, Line 230 – 231: "soil nitrogen verification was performed under the full transport condition with advection, diffusion, and dispersion."*

line 128: i think the reference is Figure S6. - same transport than previously?

**Response:** Corrected to Figure S6. Yes, same transport as the previous carbon case.

*SI, Line 230 – 231: "soil nitrogen verification was performed under the full transport condition with advection, diffusion, and dispersion."*

table S5: same comments than for table S3

**Response:** We have updated the reaction rate expression in Table S3 and S5 for consistency with microbial equation (Eqn. S9 – S11). The rate and microbial terms are described in the section of S2 (*Line 52 – 89*).

S2. what's the exponential factor?

**Response**: The exponential factor is a fitted equation (or parameters) to account for the exponentially declining rooting density at Shale Hills (Hasenmueller et al., 2017), which is also common in the forested watershed (López et al., 2001).

*Line 399 – 404: "*

$$f_{root}(d_w) = \exp((-d_w + \delta)/\lambda) \quad (14)$$

*Where $f_{root}(d_w)$ is the normalized rooting density term in the range of 0 to 1 as a function of water depth to the groundwater $(d_w)$. The rooting term (Eqn. 14) was exponentially*

*fitted ($\delta$= 0.013, $\lambda$ = 0.20) based on field measurements of root distribution along depth (Hasenmueller et al., 2017)."*

Figure S9: (a) it's really not clear what is observed. precipitation is the grey on the top, I guess. Could you write "precipitation" in gray then?
Please include evapotranspiration (ET) in the caption. The word "precipitation" in the middle of the frame is surprising.
- Figure S9 (c): there is more water in the unsaturated than in the saturated part?
**Response:** We have removed Figure S9 in SI as it does not pertain to the new model development in this paper.

---

## Author Comment (AC3) · 13 Feb 2021

It is very useful to have new models so that one can compare different approaches, especially when the models have open source code like this one. However, with very many existing catchment-scale biogeochemical water quality models, it should be more clearly stated what this model provides that others don't, and how it can shed light on catchment processes that were previously not well understood. Or alternatively how this new software makes the job of the model user easier. For instance, the process descriptions for Phosphorous, DOC and Nitrogen don't look too dissimilar to existing models. My impression is that this model is toward the complex end of the spectrum when it comes to parametric complexity. I miss an explicit analysis of this. For instance, how many parameters have to be calibrated and cannot be sufficiently constrained by (easily obtainable) measurement or literature values?

**Response:** Thanks for the comment. We have revised two introduction paragraphs to be more explicit about the existing model gap and how our integrated watershed model help understand complicated watershed processes. We have also added a paragraph for calibration process.

Line 74 – 80: *"Reactive transport models (RTMs) have traditionally centered on transport and multi-component biogeochemical reactions, typically in groundwater systems, which often have limited interactions with climate and other surficial watershed processes (Steefel et al., 2015;Li et al., 2017b;Mayer et al., 2002). Biogeochemical reactions in shallow soils that are often driven by environmental factors such as soil temperature and moisture cannot be well simulated in these models."*

Line 90 – 108: *"While many of these models can simulate reaction processes such leaching of nutrients from agriculture lands (Lindström et al., 2005;Lindström et al., 2010;Bailey et al., 2017), most of them do not explicitly solve the multi-component reactive transport equations. In other words, they have relatively crude representations of solute leaching out of element bulk mass as part of the solute export but do not represent kinetics and thermodynamics of multi-component biogeochemical reactions typically included in reactive transport models (RTMs). They also do not simulate processes such as chemical weathering. As an example, nutrient leaching is often calculated based on empirical equations without explicitly solving reactive transport equations. Reaction rates are often represented using first-order decay (Gatel et al., 2019), assuming reaction rate constants do not change with time and environmental conditions. However, biogeochemical processes including carbon decomposition and nutrient cycling are highly variable in space and time, depending on local environments such as substrate availability, soil temperature, and soil moisture (Li et al., 2017a;Suseela*

*et al., 2012;HARTLEY et al., 2007). In filling in this model need, recently we augmented our watershed model RT-Flux-PIHM (Bao et al., 2017) with new developments of microbially mediated reactions, which allows us to model the interactions between biogeochemical reactions and environmental factors that are driven by land surface and hydrological processes."*

*Line 491 – 497: "A typical model application requires 20 to 30 hydrological parameters to be calibrated. These parameters include land surface parameters (e.g., canopy resistance, surface albedo), soil and geology parameters (e.g., hydraulic conductivity, porosity, Van Genuchten, macropore properties) (Shi et al., 2013). Reaction-related parameters (e.g., reaction rate constant, mineral surface area, $Q_{10}$, $S_{w,c}$, and n) are additionally needed for calibration, the number of which depends on the numbers of reactions involved in a particular system."*

Knowledge of this is of vital importance to a user. For instance, if one wants to do a very thorough investigation of the processes in a single catchment one maybe has some time to spend to do a very detailed model setup. However, in some applications one needs to model all the inputs from land into a whole coastline or a large set of lakes. In such an application one often relies on autocalibration and upscaling, and in such applications high parametric complexity can be detrimental. On a similar note, data availability of data that can be used as model drivers can vary between locations. Does this model accommodate for locations with low data availability?

**Response:** Thanks for the interesting point. The short answer for low data availability is mostly YES as our model can be operated on the spatially lumped version, which requires much less data for spatial details and data points in different locations. And the model is flexible and can take inputs either from online data portals or user's own measurements (e.g., elevation, soil properties). The model uses a global coefficient approach to reduce parameter dimension and facilitate parameter calibration (*Line 490 – 491*).

*Line 197 – 200: "Despite the model complexity, the model is flexible for taking inputs from online data portals or local measurements and it can accommodate low data availability (see the following section of 5 for data need and domain setup)."*

*Line 445 – 447: "the model domain can be set up using elevation, land cover, soil and geology maps supplied by the user or from the data portal of Geospatial Data Gateway (https://datagateway.nrcs.usda.gov)."*

*Line 455 – 456: "Local measurements from meteorological stations and field campaigns (e.g., land cover, soil, geology) can also be used in the model."*

*Line 481 – 483: "A simple domain can be set up with only two land grids representing two sides of a watershed connected by one river cell. This setup uses averaged properties without needs for larger spatial data."*

*Line 490 – 491: "Auto-calibration is not built into the model, but a global calibration coefficient approach is used to reduce parameter dimension and facilitate manual calibration."*

If the stated goal of the model is to be a research model targeted at understanding catchment processes, rather than a model that can also be used as an input source for oceanic models or to be used by government officials to inform policy decisions on a large scale, then this is maybe not as big of a concern. But that could be made more explicit.
**Response:** We respectfully disagree. How to use the model should be determined by users, not model developers. For example, HBV is a simple hydrology model but it has been used by many in the world for policy making and guidelines for decisions. Users should decide how they would like to use the code.

Can you argue why the model complexity is justified? Some studies show that simple models can give as good predictions as complex ones while taking much less time and data to deploy. I can see that you have a plot of sensitivity to turning off various nitrate processes, but what about sensitivity to simpler or more complex descriptions of these processes? What is the sensitivity of model results to perturbations in the parameters?
**Response:** This is a generic question to the whole modeling community, not just about this manuscript. Nonetheless, we added some discussions on model simplicity and complexity. In a nutshell, we advocate for simple models and adding complexity only if necessary. We added the following:

*Line 723 – 756: "The model presented here is complex and process-based. The computational cost of solving a spatially distributed, nonlinear, multi-component reactive transport model is high, posing challenges for the application of ensemble-based uncertainty analysis and model weighting/selection methods (Song et al., 2015). With additional reaction and transport processes, the model includes more functions (such as reaction kinetic rate laws) and parameters (e.g., reaction rate constants, surface area) than hydrological models, which have already been criticized for their complexity, equifinality, uncertainty, and data demands (Beven, 2001, 2006;Kirchner et al., 1996). These issues will persist even though reactive transport models will be constrained by additional chemical data. A major source of uncertainty in these models lies in epistemic uncertainties, i.e., the lack of specific knowledge in forcing data and details of reactivities (e.g., spatial distribution and abundance of reactive materials), on top of uncertainties related to hydrology (Beven, 2000;Beven and Freer, 2001). The model's conceptual foundations also represent a major source of uncertainty.*

*It is in this spirit of "balancing" the cost and gain that we present both spatial distributed and lumped modes for the BioRT model. Compared to the distributed version, the spatially implicit model requires less spatial data, is computationally inexpensive, and*

*is relatively easy to set up. It can assess the average dynamics of the water and solute dynamics and focus on the interactions among processes without resolving spatial details. The lumped approach can also accommodate basins with low data availability, and it can be easier for students to learn to use the model. In contrast, spatially explicit representations enable the exploration of the "hot spots" (e.g., swales and riparian zones with high soil water DOC concentrations in Figure 10e) and their contribution to stream chemistry at different times. Spatial heterogeneities in watershed properties (e.g., soil types and depth, lithology, vegetation, biomass, and mineralogy) are ubiquitous in natural systems. However, a general understanding of the linkage between local catchment features and catchment-scale dynamics (e.g., stream concentration dynamics and solute export pattern) is often lacking. We generally do not understand how spatial heterogeneity affects water flow paths, stream water chemistry, and biogeochemical reaction rates. The spatially distributed model provides a tool to further explore these questions. Ultimately, the choice of the model complexity level depends on the research questions that the model is set to answer. At the end, we all need to balance cost and gain when deciding to use a simple or complex model, striving to be "simple but not simplistic" (Beven and Lane, 2019)."*

Similarly, what is the sensitivity to subdividing the land into many cells? Is having 100 cells warranted, or can you get just as good predictions just using a couple of cells describing the different land use types? (I understand the argument about identifying hot spots, but it could also be interesting to see if the subdivision has impact on the stream concentration predictions).

**Response:** This is an interesting upscaling-related question that deserves its own manuscript. With lumped model, one gets the simplicity and average dynamics and bound to lose spatial details. Our discussion above provides some guidelines. But the other two reviewers are also saying this paper is too long and has too much information so we are not adding more here. It would be interesting to have an independent paper asking just this upscaling question.

What is the calibration process like for the user? Are any autocalibration tools set up for the model?

**Response:** This is a good point. We have added a few sentences to describe the calibration part.

*Line 490 – 497: "Auto-calibration is not built into the model, but a global calibration coefficient approach is used to reduce parameter dimension and facilitate manual calibration. A typical model application requires 20 to 30 hydrological parameters to be calibrated. These parameters include land surface parameters (e.g., canopy resistance, surface albedo), soil and geology parameters (e.g., hydraulic conductivity, porosity, Van*

*Genuchten, macropore properties) (Shi et al., 2013). Reaction-related parameters (e.g., reaction rate constant, mineral surface area, $Q_{10}$, $S_{w,c}$, and n) are additionally needed for calibration, the number of which depends on the numbers of reactions involved in a particular system."*

---

## Author Response (AR2)

**Comment on "BioRT-Flux-PIHM v1.0: a watershed biogeochemical reactive transport model" by Wei Zhi et al.**

**Topical Editor decision**

Received: 21 Sep 2021

Dear Editor Min-Hui Lo:

Thank you for your speedy handling of our manuscript. We have done another round of editing, including:
1) Removed some outdated references and updated with some new references.
2) Removed some redundancy in the text.
3) Added some explanations (e.g., Nash Sutcliffe efficiency, what kind of spatial data) for clarity.
4) Added a Table S7 and a brief summary for model validation performance as suggested.
5) Added a Zenodo DOI in the section of Code availability.

1. Although the authors have mentioned the verification of the BioRT model in the revised manuscript (The BioRT module had been verified against CrunchTope under different transport and reaction conditions (Figures S1 – S7 in SI), the performance and verification of the model can be further improved. Especially, this is a Model description paper, so it will be helpful for the readers and users if the authors can provide some statistics in the main text (maybe at the discussion section?) for the performance of BioRT compared to CrunchTope (maybe a summary table from those materials in SI?)

**Response:** Thank you for the comments. We have added model performance statistics in the section of 4. Numerical scheme and model verification (Line 394 – 397) and a summary Table S7 in the section of S3.4 Validation performance summary.

Line 394 – 397: "Table S7 shows an average percent bias and Nash Sutcliffe efficiency (NSE) of 1.1% and 0.98, indicating a robust performance for a variety of solutes under different transport and reaction conditions."

S3.4. Validation performance summary

Model validation performance for above-mentioned cases using percent bias (PBIAS) and Nash Sutcliffe efficiency (NSE) is summarized in Table S7. The optimal

value of PBIAS is 0, with low-magnitude values indicating accurate model simulation (Moriasi et al., 2007). Positive values indicate model underestimation bias, and negative values indicate model overestimation bias. NSE ranges between $-\infty$ and 1, with NSE = 1 being the perfect fit (Moriasi et al., 2007).

**Table S7.** Model validation performance

| Process | Transport | Species | PBIAS (%) | NSE |
|---|---|---|---|---|
| Phosphorus (Fig S2, S3) | Advection-only | $H^+$ | 3.1 | 0.96 |
| | | $Cl^-$ | 1.0 | 0.99 |
| | | TP | 1.0 | 0.99 |
| | | $HPO_4^{2-}$ | -2.3 | 0.99 |
| | | $H_2PO_4^-$ | 1.7 | 0.99 |
| | | $H_3PO_4$ | 4.6 | 0.95 |
| | Advection + diffusion + dispersion | H+ | 2.7 | 0.97 |
| | | $Cl^-$ | -0.27 | 1.0 |
| | | TP | -0.20 | 1.0 |
| | | $HPO_4^{2-}$ | -3.2 | 0.98 |
| | | $H_2PO_4^-$ | 1.4 | 0.99 |
| | | $H_3PO_4$ | 4.5 | 0.96 |
| Carbon (Fig S5) | Advection + diffusion + dispersion | $O_2(aq)$ | 2.2 | 0.98 |
| | | $NO_3^-$ | -1.4 | 0.99 |
| | | $SO_4^{2-}$ | -0.2 | 0.99 |
| | | $HCO_3^-$ | 1.1 | 0.99 |
| | | $N_2(aq)$ | 2.0 | 0.98 |
| | | $H_2S(aq)$ | 2.5 | 0.98 |
| Nitrogen (Fig S7) | Advection + diffusion + dispersion | $O_2(aq)$ | 2.2 | 0.99 |
| | | NH4+ | 1.5 | 1.0 |
| | | NO3- | -1.3 | 0.98 |
| | | N2(aq) | 1.8 | 0.99 |

2. Please remove the reference from the abstract (GMD - Submission (geoscientific-model-development.net)).
**Response:** Removed as suggested. Thanks.

3. Please provide the doi from zenodo for the data and code section. (https://www.geoscientific-model-development.net/policies/code_and_data_policy.html)
**Response:** The DOI from Zenodo has been added to the code section (Line 739 – 742).

Line 739 – 742: "Code availability. The current model release (BioRT-Flux-PIHM v1.0) is archived at: https://doi.org/10.5281/zenodo.3936073. Documentation, source code, and examples are available at GitHub repository: https://github.com/Li-Reactive-Water-Group/BioRT-Flux-PIHM.

4. The authors have responded to original reviewer #2 that the Nash Sutcliffe efficiency has been removed. However, the revised manuscript L660-661 (The model outputs followed the general trend of stream DOC measurements (NSE = 0.55 for monthly DOC) still has NSE with only the abbreviation. Would you please address this carefully?
**Response:** Thanks for the catch. We have defined NSE on the first appearance (Line 395) and added an explanation in the text on Line 618 – 622.

Line 618 – 622: "The model outputs followed the general trend of stream DOC measurements with the model evaluation index NSE of 0.55 for monthly DOC concentration (Figure 10a). NSE ranges from $-\infty$ to 1.0 (i.e., perfect fit) with values greater than 0.5 considered good performance for monthly water quality model (Moriasi et al., 2015) "

5. L487: "It requires much more data and can be computationally expensive but can be used to identify "hot spots" of biogeochemical reactions within a watershed". What kind of data? Please be more specific.
**Response:** Thanks for the comment. We have revised the paragraph and added some explanations for spatial data on Line 443 – 446.

Line 443 – 446: "Alternatively, a complex domain can be set up to track "hot spots" of biogeochemical reactions using many grids with explicit representation of spatial details (e.g., topographic map, river network, land use map, soil and geology map, mineral distribution)."

**Reference:**
Moriasi, D. N., Arnold, J. G., Van Liew, M. W., Bingner, R. L., Harmel, R. D., and Veith, T. L.: Model evaluation guidelines for systematic quantification of accuracy in watershed simulations, T Asabe, 50, 885-900, 2007.
Moriasi, D. N., Gitau, M. W., Pai, N., and Daggupati, P.: Hydrologic and water quality models: Performance measures and evaluation criteria, T Asabe, 58, 1763-1785, 2015.

---

## Author Response (AR3)

**Comment on "BioRT-Flux-PIHM v1.0: a watershed biogeochemical reactive transport model" by Wei Zhi et al.**

**Topical Editor decision**

Received: 6 Oct 2021

Dear Editor Min-Hui Lo:

Thank you for your speedy handling of our manuscript. We have made minor changes as suggested, as below:

1) Thanks for further illustrating the meaning of NSE (i.e., NSE rangers between -infinity to 1, with NSE = 1 being the perfect fit (Moriasi et al., 2007)); however, can you move to the 1st time when you introduce NSE index?
Response: moved to the first mention of the NSE index as suggested. Thanks.

Line 398 – 399: "Note NSE ranges from $-\infty$ to 1, with NSE = 1 being the perfect fit (Moriasi et al., 2007)."

2) Please also provide the doi from zenodo for "Documentation, source code, and examples", which now is only from the github link: https://github.com/Li-Reactive-Water-Group/BioRT-741 Flux-PIHM.
Response: we have added a Zenodo link in the text as suggested.

Line 114 – 117: "The source code and the examples shown here are archived on the Zenodo website (https://doi.org/10.5281/zenodo.3936073) and the GitHub website (https://github.com/Li-Reactive-Water-Group/BioRT-Flux-PIHM)."

Line 739 – 742: "**Code availability**. The current model release (BioRT-Flux-PIHM v1.0) is archived at: https://doi.org/10.5281/zenodo.3936073. Documentation, source code, and examples are available at GitHub repository: https://github.com/Li-Reactive-Water-Group/BioRT-Flux-PIHM."